

# Methane Pathways in Winter Ice of Thermokarst Lakes, Lagoons and Coastal Waters in North Siberia

Ines Spangenberg[2,1], Pier Paul Overduin[2], Ellen Damm[2], Ingeborg Bussmann[3], Hanno Meyer[2], Susanne Liebner[4,1], Michael Angelopoulos[2], Boris K. Biskaborn[2], Mikhail N. Grigoriev[5], and Guido Grosse[2,1]

[1]Institute of Geosciences, University of Potsdam, Potsdam, Germany
[2]Alfred Wegener Institute Helmholtz Centre for Polar and Marine Research, Potsdam, Germany
[3]Alfred Wegener Institute Helmholtz Centre for Polar and Marine Research, Helgoland, Germany
[4]Section of Geomicrobiology, Helmholtz Centre Potsdam - GFZ German Research Centre for Geosciences, Potsdam, Germany
[5]Mel'nikov Permafrost Institute, Siberian Branch, Russian Academy of Sciences, Yakutsk, Russia

**Correspondence:** Paul Overduin (paul.overduin@awi.de)

**Abstract.** The thermokarst lakes of permafrost regions play a major role in the global carbon cycle. These lakes are sources of methane to the atmosphere but the methane flux is restricted by an ice cover for most of the year. The fate of methane in these waters and is poorly understood. We provide insights into the methane pathways in the winter ice cover on three different water bodies in a continuous permafrost region in Siberia. The first is a bay underlain by submarine permafrost

(Tiksi Bay, TB), the second a shallow thermokarst lagoon (Polar Fox, PF) and the third a land-locked, freshwater thermokarst lake (Goltsovoye Lake, GL). In total, 11 ice cores were analyzed as records of the freezing process and methane pathways during the winter season. In TB, the hydrochemical parameters indicate an open system freezing. In contrast, PF was classified as a semi-closed system, where ice growth eventually cuts off exchange between the lagoon and the ocean. The GL is a closed system without connections to other water bodies. Ice on all water bodies was mostly methane-supersaturated with

respect to the atmospheric equilibrium concentration, except of three cores from the lake. Generally, the TB ice had low methane concentrations (3.48–8.44 nM) compared to maximum concentrations of the PF ice (2.59-539 nM) and widely varying concentrations in the GL ice (0.02–14817 nM). Stable $\delta^{13}C_{CH_4}$ isotope signatures indicate that methane above the ice-water interface was oxidized to concentrations close to or below the calculated atmospheric equilibrium concentration in the ice of PF. We conclude that methane oxidation in ice may decrease methane concentrations during winter. Therefore, understanding

seasonal effects to methane pathways in Arctic saline influenced or freshwater systems is critical to anticipate permafrost carbon feedbacks in course of global warming.

Keywords for the paper: Permafrost, lake ice, methane pathways, carbon fluxes, stable isotopes....

## 1 Introduction

Methane ($CH_4$) is, after water vapor ($H_2O$) and carbon dioxide ($CO_2$), the most abundant greenhouse gas in the troposphere

(Wuebbles and Hayhoe, 2002). When averaged over a 100 year timescale, the warming effect of methane per unit mass is 28



times higher than that of carbon dioxide (Stocker, 2014). As a large reservoir of organic carbon, permafrost holds a potential positive feedback to climate warming (Strauss et al., 2013; Hugelius et al., 2014; Schuur et al., 2015). Amplified warming of the atmosphere in the Arctic is causing substantial increases of ground temperatures and associated thawing of permafrost (Biskaborn et al., 2019). As a consequence organic carbon stored in permafrost becomes available for microbial decomposition,

leading to the release of greenhouse gases (Koven et al., 2011; Knoblauch et al., 2018). Therefore, these regions play an important role in the global carbon cycle and contribute to global warming (Schuur et al., 2015; Turetsky et al., 2019).

Arctic aquatic systems represent essential components for carbon cycling in permafrost regions, as under certain circumstances i.e. insufficiently deep waters, methane can be produced over the whole year compared to active layer soils that thaw in summer only (Wik et al., 2016). In summer, methane is produced in anaerobic parts of unfrozen permafrost soils and sediments.

However, as most Arctic waters are covered with ice for 9-10 months, winter methane emissions from water bodies in the Arctic region are strongly reduced. During winter, the active layer is frozen and methane production is limited to unfrozen taliks (a layer of unfrozen soil above the permafrost and below the water). Waters that are deep enough to not completely freeze to the bottom in the winter are underlain by taliks (Arp et al., 2016) and may have particularly high methane emissions if organic-rich permafrost deposits underneath are actively thawing and microbial decomposition results in methane production (Walter et al.,

2007). Recent studies have illustrated the importance of the winter period for annual methane and $CO_2$ emission budgets of northern lakes (Denfeld et al., 2018; Powers and Hampton, 2016; Zimov et al., 2006); however, processes controlling methane and $CO_2$ dynamics in the frozen period are still poorly understood. Generally, the amount of methane emitted from Arctic water bodies to the atmosphere is uncertain but is expected to increase as a result of arctic warming (Bastviken et al., 2004; Boereboom et al., 2012; Cole et al., 2007; Walter et al., 2006). Since seasonally ice-covered lakes dominate permafrost land-

scapes, they are expected to play a major role in the global carbon cycle (Grosse et al., 2013; Wik et al., 2016; Bartsch et al., 2017).

Northern high latitude lakes, i.e. thermokarst lakes, are suggested to act as methane sources throughout the entire annual cycle (Wik et al., 2016). Methane is produced mainly in carbon-bearing sediments under anoxic conditions (e.g. Conrad, 2009). Methane is transported from sediment layers into the water column by processes described as diffusion and ebulli-

tion (Bastviken et al., 2008). The production of methane is temperature-dependent (Kelly and Chynoweth, 1981; Zeikus and Winfrey, 1976), but methane ebullition and accumulation under ice suggest methane production continues during winter (Walter Anthony et al., 2010). In particular in first generation thermokarst lakes, are generally ice- and carbon rich permafrost thaws for the first time, a large amount of methane migrates as bubbles from the sediments into the water column (Zimov et al., 2006). While gas may easily escape to the atmosphere in summer, an ice cover forms a barrier in winter and traps bubbles under and

eventually within the ice (Walter et al., 2008). Methane may be oxidized at the ice-water interface (Canelhas et al., 2016; Rudd and Hamilton, 1978), but the conditions required are poorly understood (Canelhas et al., 2016).

Previous work has focused mostly on the occurrence and spatial variability of methane bubbles within lake ice (Walter et al., 2006; Wik et al., 2011; Sasaki et al., 2009; Walter Anthony et al., 2010) and $CH_4$ accumulation underneath lake ice (Boereboom et al., 2012; Langer et al., 2015). Remote sensing was effectively used for detecting the distribution of methane

bubbles in thermokarst lake ice from Alaska (Lindgren et al., 2016). However, few studies have focused on the fate of methane





accumulation in the ice cover and at the ice-water interface during winter. Methane distribution in ice cores and under ice were analysed in shallow ice-covered tundra lakes in Alaska (Phelps et al., 1998). Boereboom et al. (2012) examined $CO_2$ and methane in the ice covering four lakes in a discontinuous permafrost area. In both studies the accumulation of methane in and under the ice during winter was investigated. Their findings are supported by other studies, which also show a high

methane emission rate during ice melt in spring of high latitude lakes (Karlsson et al., 2013; Michmerhuizen et al., 1996). Methane oxidation observed in Arctic shelf waters (Damm et al., 2005; Mau et al., 2013; Bussmann et al., 2017), and sediments (Overduin et al., 2015; Winkel et al., 2018), as well as in lake water (Bastviken et al., 2002; Lidstrom and Somers, 1984), and at the water-ice interface (Martinez-Cruz et al., 2015; Canelhas et al., 2016) may limit the emissions of methane during spring ice melt (Canelhas et al., 2016). While Phelps et al. (1998) suggest that methanotrophic activity is generally inhibited in cold

waters, methane is still being oxidized at temperatures as low as 2°C (Canelhas et al., 2016).

Only a few studies compare ice from different types of Arctic water bodies, crossing freshwater to saline gradients. However, the transition from lakes to lagoons and further to bay waters via sea level rise and permafrost coastal erosion processes (e.g. Romanovskii et al., 2000) may substantially change the methane budget of aquatic systems in permafrost regions. Methane fluxes in lake systems are coupled to the lake's limnological and geomorphological characteristics (Bastviken et al., 2008,

2004; Boereboom et al., 2012; Denfeld et al., 2018). Studies of water bodies covering the different stages of a lake-lagoon-shelf transition allow us to compare their changing methane fluxes in rapidly changing aquatic systems of permafrost regions and to improve our understanding of how water bodies function as methane sources or sinks.

This study aims to clarify the role of an winter ice cover for methane cycles of three different stages in the lake-lagoon-shelf transition in a region of rapidly thawing permafrost in northeast Siberia. Our objective is to demonstrate how methane is

distributed within seasonal ice from Tiksi bay (TB), Polar Fox Lagoon (PF), and Goltsovoye Lake (GL), to better understand 1) how freezing processes differ between the three water bodies, 2) what the relationships between freezing dynamics and methane concentration in the ice are, and 3) the potential importance of methane oxidation in different water bodies.

## 2   Study Area

This study was conducted on the southern coast of Bykovsky Peninsula at 71° 40' - 71° 80' N and 129° 00' - 129° 30' E,

in northeast Siberia. Bykovsky Peninsula is located in the continuous permafrost zone, north-east of the harbor town of Tiksi and approximately 20 km southeast of the Lena River Delta, central Laptev Sea (Grosse et al., 2007). This area offers a rich variety of water body types in a continuous permafrost setting with ice-rich Yedoma permafrost, a widespread, organic-rich and syngenetic fine-grained deposit that accumulated during the glacial age in the unglaciated regions of Siberia and Central Alaska (Zimov et al., 2006; Schirrmeister et al., 2011). Most thermokarst lakes in that area originated in the early Holocene, when

surficial permafrost thawed. Their sediments typically have total organic carbon (TOC) contents of about 5-30 % by weight (Biskaborn et al., 2012, 2013a, b, 2016; Schleusner et al., 2015). Preliminary measurements of TOC in the lake sediments of GL and PF showed values of up to 9-10 wt% (unpublished data). The coastline erodes at mean rates of between 0.5 and 2 m per year and can intersect these water bodies, draining them or leading to the formation of thermokarst lagoons (Lantuit et al.,

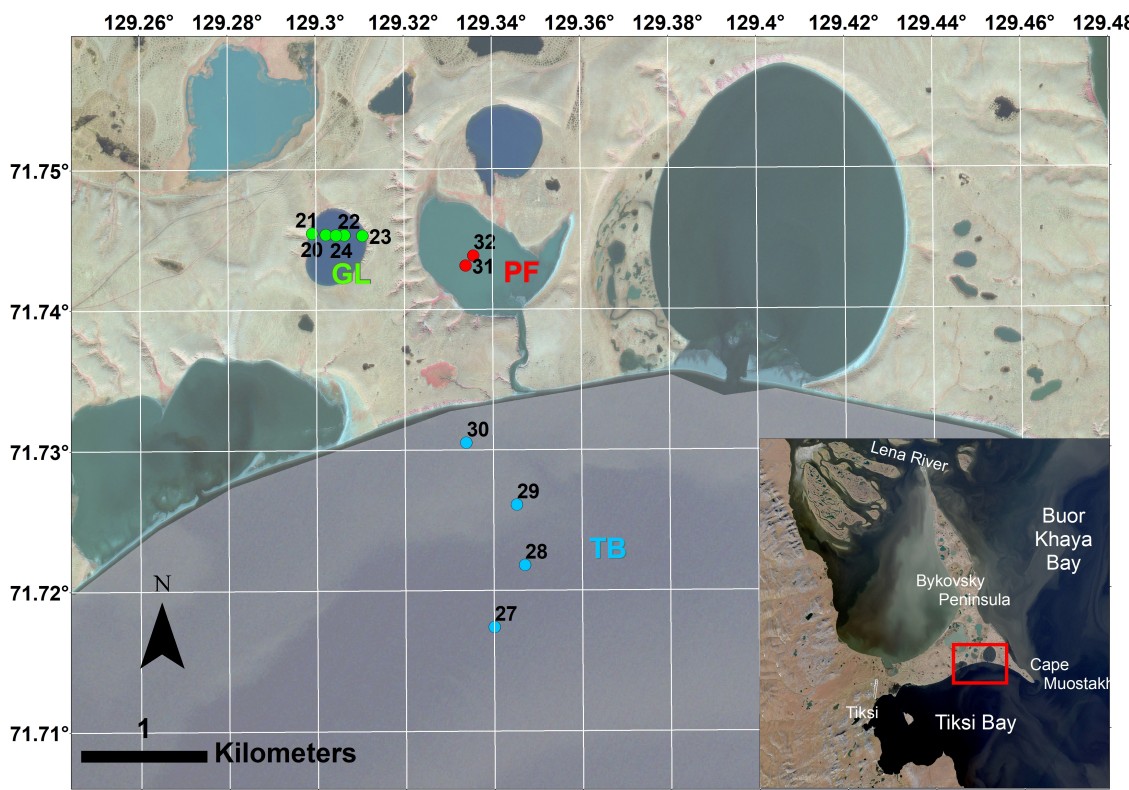

**Figure 1.** Study design on the Bykovsky Peninsula, NE Yakutia, Arctic Russia. Ice cores were recovered from three water bodies on the southern shore of Bykovsky Peninsula. Ice cores were recovered from Tiksi Bay (TB: blue), Polar Fox Lagoon (PF: red), and Goltsovoye Lake (GL: green dots). Imagery in the main map is Worldview-3 from Sep. 9, 2016 ((c) DigitalGlobe).

2011). For our study, we selected three water bodies at the southern coast of Bykovsky Peninsula: Tiksi Bay (TB), Polar Fox Lagoon (PF) and Goltsovoye Lake (GL) (Fig. 1, Tab. 1).

TB is a relatively shallow bay underlain by submarine permafrost (Overduin et al., 2016), at least close to the coast. The bay is located south-east of the Bykovskaya Channel, which is the major outlet of the Lena River through which about 25 % of the Lena's spring discharge exits the delta (Fedorova et al., 2015). The mixing of freshwater from the Lena River with saline water from the Laptev Sea results in brackish conditions in the bay (Lantuit et al., 2011). TB belongs to the Buor Khaya Bay region, where the water column is usually stratified, with a colder, more saline water layer underlying the brackish surface layer (Overduin et al., 2016). The depth of the pycnocline varies between 4 and 10 m and the stratification can be disturbed by storm events. Storm surges also influence the sea level at TB, with maximum wave heights of about 1.1 m. Tidally-based sea-level oscillations have little influence on the height of storm surges and the Bykovsky Peninsula region is characterized by a micro-tidal regime (Lantuit et al., 2011). PF is connected to TB by a shallow channel (Fig. 1) and therefore dominated by brackish water. PF represents a transitional stage between a land-bound thermokarst lake and a bay: a thermokarst lagoon. Its morphology suggests that it has been formed in a thermokarst depression. The lower extent of ice-rich sediments and the





thermokarst lake beds lie below sea level (Günther et al., 2015). GL is a slightly oval-shaped thermokarst lake about $0.5\,\mathrm{km}$ in diameter, surrounded by Yedoma uplands at various stages of degradation. Tab. 1 lists description of the size and maximum depths of the studied water bodies.

**Table 1.** Hydrological characteristics of water bodies of the southern Bykovsky Peninsula from which ice cores were taken for this study in spring 2017 (Strauss et al., 2018). Temperatures and electrical conductivity for GL and PF were measured in the field below the ice; temperature and salinity for TB are from Charkin et al. (2017).

| Water Body | Diameter [km] | Max. Depth [m] | Water temperature [°C] | Electrical Conductivity | Number of Ice Cores | Mean Ice Thickness [cm] |
|---|---|---|---|---|---|---|
| Tiksi Bay (**TB**) | $10 - 26$ | 11 | $-1$ to $0$ | 8.5–12.5 PSU | 4 | 144 |
| Polar Fox Lagoon (**PF**) | $0.61 - 1.1$ | 4 | $-0.75$ | $11\,\mathrm{mS/cm}$ | 2 | 166 |
| Goltsovoye Lake (**GL**) | $0.46 - 0.62$ | 10 | $0$ to $1.8$ | $0.3\,\mathrm{mS/cm}$ | 5 | 160 |

## 3 Methods

### 3.1 Sampling in the field

Ice cores were collected from TB in a transect roughly perpendicular to the shore (Fig. 1). For PF, two cores were drilled near the center of the lagoon. The sites of the five cores from GL were located along an approximately east-west transect across the lake. Tab. 1 lists the mean ice thicknesses of the sampled ice core for the locations.

The ice cores were taken with a Kovacs Mark II ice coring system ($9\,\mathrm{cm}$ diameter), between Apr. 5 and 12, 2017. Cores were collected in triplicate from each sampling site. One core was used for temperature measurements, one was collected for genetic studies, and the third was used for hydrochemical, stable isotope and methane measurements. Immediately after sampling, temperature was measured by drilling small holes every $10\,\mathrm{cm}$, at the boundaries between sample sections, into one of the cores and inserting a digital thermometer. Average temperature values were calculated for each section from the top and bottom depths. The top ($10\,\mathrm{cm}$), middle ($10\,\mathrm{cm}$) and lower ($30\,\mathrm{cm}$) part of the ice cores (except for core 32) were used for another study and available for temperature and stable water isotope data only. res were wrapped in sealed plastic tubes and packed in thermally insulated boxes for the transport to Germany. Water column temperature and electrical conductivity were measured with a CTD (SonTek CastAway) at each coring site. Further details on field work and sampling can be found in the expedition report (Strauss et al., 2018).



## 3.2 Sample processing

The ice cores were processed in Potsdam from Dec. 4 to 15, 2017 (Cores 23, 24, 27, 28, 29, 30) and from the Apr. 30 to Mar. 4, 2018 (Cores 20, 21, 22). The ice cores were cut in a cold room at $-15\,°C$ with a band saw every $10\,cm$ and stored for melting in gas-tight TEDLAR bags at $4$ to $7\,°C$ (over 1-2 days). The closed bags were evacuated with a vacuum pump before.

After melting, the bags were gently mixed and water was poured through tubing, without producing bubbles, into $100\,mL$ glass bottles for the analysis of $CH_4$ concentrations and $\delta^{13}C$ in $CH_4$. The remaining water was distributed into other sample bottles for hydrochemical measurements of pH, electrical conductivity (EC), dissolved organic carbon (DOC), $\delta^{18}O$ and $\delta D$ isotopes of water, as well as major anion and cation concentrations. Data points in the graphs of variation against core depth thus represent a mean value for a section of usually $10\,cm$ within an ice core.

## 10   3.3 Hydrochemistry in ice

Electrical conductivity and temperature of GL and PF were measured in the field. EC and pH of ice core samples were measured with a WTW Multilab 540 device as soon as possible after bottling. The salinity was calculated from the values of the electrical conductivity after McDougall and Barker (2011). The samples for DOC were filtered with $0.7\,\mu m$ pore size glass fiber filters (the filters were first rinsed with $20\,mL$ of the sample), filled in $20\,mL$ glass-headspace vials and closed with aluminum

crimp caps. For preservation, $50\,\mu L$ of 30 % supra-pure HCl were added to the sample before closing the vials, which were stored at $4\,°C$ until measurement. DOC was measured with a Shimadzu Total Organic Carbon Analyzer (TOC-VCPH). An average of three to five injections per sample was used as the measured value. The detection limit for the DOC measurement is $0.25\,mg\,L^{-1}$ and the uncertainty of the measurement was ±10 % for values higher than $1.5\,mg\,L^{-1}$, and for values lower than $1.5\,mg\,L^{-1}$ the uncertainty was ±15-20 %.

## 20   3.4 Stable water isotopes

To measure stable water isotopes ($\delta^{18}O$, $\delta D$), $10\,mL$ of the untreated water sample was filled in $10\,mL$ PE narrow-neck bottles. Samples with salinity higher $300\,\mu S\,cm^{-1}$ were measured with an Isotope Ratio Mass Spectrometer (IRMS: Finnigan Delta-S), using equilibration techniques (Meyer et al., 2000). Whereas samples with low salinity were measured with an Ultra High-Precision Isotopic Water Analyzer (PICARRO L2130-i, coupled with an autosampler and vaporizer using Cavity

Ringdown Spectroscopy). The internal precision of the H and O isotope measurements is better than ±0.8 ‰ and ±0.10 ‰, respectively. The oxygen and hydrogen isotopic compositions are given relative to Standard Mean Ocean Water (VSMOW) using the conventional $\delta$-notation.

Stable water isotopes have been widely used in palaeoclimate and palaeohydrological research as isotope fractionation is temperature-dependent. The mean annual $\delta^{18}O$ of precipitation is positively correlated with the mean annual air temperature,

and hence snow typically has a strong isotope variability as well as relatively low (or light) values, particularly at high latitudes (Dansgaard, 1964). Stable water isotopes can be also used to trace water phase changes i.e. during freezing as these are accompanied by kinetic isotope fractionation processes (Souchez and Jouzel, 1984; Lacelle, 2011). An $\delta^{18}O$-$\delta D$ plot gives





valuable information about the freezing characteristics, as the regression line (freezing line) is usually lower than that of the Global Meteoric Water Line (GMWL, slope = 8). S lope values in the range of 6 to 7.3 can be interpreted as freezing under equilibrium conditions (Lacelle, 2011). The extent of fractionation in the system water-ice critically depends on the velocity and rate of freezing (Gibson and Prowse, 1999; Tranter, 2011) which in turn is directly connected to the thermal conditions and the water availability of a given system. Isotopic fractionation during freezing is accompanied by heavier isotope composition for the first ice and lighter isotope composition for the last ice formed (Gibson and Prowse, 1999, 2002). In this study, we differentiate between open and closed system freezing. In an open system, the water source under the ice and hence the isotope composition of the ice formed both remain largely constant, differing from closed-system freezing where the isotopic composition of the water pool changes prior to freezing. Furthermore, the water isotopic signature may be indicative for the mixing of different water masses (endmembers) i.e. precipitation with surface water. The isotopic signature is then indicative for the relative contribution of each endmember and preceding natural isotopic fractionation processes, which changes if the endmembers' specific isotope composition differs. The $\delta^{18}$O-$\delta$D values in ice may also be directly influenced by precipitation, if liquid or solid precipitation falls on the ice layer and freezes to a part of the ice. In a co-isotope plot, $\delta^{18}$O and $\delta$D of precipitation generally have values that lie on or near to the Global Meteoric Water Line (GMWL) (Craig, 1961).

## 3.5 Dissolved methane concentration

Meltwater from the TEDLAR bags was filled until overflowing into $100\,\text{mL}$ glass bottles, sealed with butyl stoppers and crimped with aluminum plugs. The samples were kept cold ($4\,°\text{C}$) and dark until the measurements in the lab (max. 2 months between sampling and measurement). Methane concentrations were measured from Jan. 29–Feb. 21, 2018 (cores 23, 24, 27–30) and on Jun. 21–Jul. 4, 2018 (cores 20–22). For methane concentration, $5\,\text{mL}$ of $N_2$ was added into the vials, and then equilibrated for 1 hour at room temperature. Normally, $1.5\,\text{mL}$ of a sample was injected into a gas chromatograph (GC; Agilent 8900) with a flame ionization detector (FID). For gas chromatographic separation, a packed column (Porapac Q 80/100 mesh) was used. The GC was operated isothermally ($60\,°\text{C}$) and the FID was held at $250\,°\text{C}$. Standards of different gas mixtures between 1.665 and 100 ppm were used for calibration, yielded standard deviations of <0.3 ppm. The GC precision had an error of 1 %. 46 % (n = 74) of the samples had methane concentrations below the lowest standard of 1.665 ppm (ranging from 0.7 ppm to 1.6 ppm), while 11 % (n = 18) had concentrations exceeding the highest standard (ranging from 105 ppm to 2723 ppm).

## 3.6 Carbon isotopes of methane

The carbon isotopic composition of methane ($\delta^{13}C_{CH_4}$) was measured on the same day and at the same bottle as the methane concentrations, to assure comparability of the data. After measuring the methane concentration, $20\,\text{mL}$ of $N_2$ were added to the sample bottle to increase the headspace of the bottle for stable carbon isotope measurements. The bottle was shaken for at least 30 minutes. $20\,\text{mL}$ of gas were removed with a glass syringe by adding $20\,\text{mL}$ of Milli-Q water at the same time to equilibrate pressure. $\delta^{13}C_{CH_4}$ was determined using a Delta XP plus Finnigan mass spectrometer. The extracted gas was purged and trapped with PreCon equipment (Finnigan) to pre-concentrate the sample. The carbon isotopic ratios are given relative to the Vienna Pee Dee Belemnite (VPDB) standard using the conventional $\delta$-notation. The analytical error of the analyses is ±1.5





‰ for $\delta^{13}C_{CH_4}$ values. Methane concentration in nanomolar (nM) was calculated with the Bunsen solubility coefficient of Wiesenburg and Guinasso Jr (1979).

For all water bodies, a Rayleigh distillation model of the type discussed by Coleman et al. (1981) and used by Damm et al. (2005, 2015) was calculated:

$$\delta^{13}C_{CH_4} = 1000 \left( \frac{1}{\alpha} - 1 \right) \ln f + \left( \delta^{13}C_{CH_4} \right)_0 \tag{1}$$

where $\alpha$ is the isotope fractionation factor, $f$ is the fraction of the methane remaining and $(\delta^{13}C_{CH_4})_0$ is the initial isotopic composition. For the Rayleigh model, bacterial oxidation was assumed to be the only methane sink, with no further inputs or mixing that would affect the isotopic composition of methane (Mook, 1994).

### 3.7 Bubble transect

To gain insight into the type and spatial distribution of methane bubbles trapped in the ice of GL a methane bubble transect was mapped. Snow was cleared from a lake ice transect of $70\,m$ length and $1.4\,m$ width starting $30\,m$ from the northwest shoreline aiming towards the lake centre (Fig. 6). A GoPro Hero 7 camera was used to take densely overlapping photos of the transect from approx. $1.7\,m$ vertical height along the entire transect to photographically record the methane bubble patches captured in the lake ice that are associated with certain seeps types. A measurement tape at the side of the ice-free transect area served as scale in the images. The photos were rectified and mosaicked in the image rectification software AGISoft Professional and the georeferenced mosaic was imported to a desktop geographical information system (ArcGIS, version 10.4). Methane seep types were classified and mapped following Walter Anthony and Anthony (2013). The distance to the shoreline of seep classes was calculated in ArcGIS using the "near" function. The Kernel density estimation in 14 distance classes was calculated in the R environment.

### 4 Results

#### 4.1 Ice morphology

We compared the data within ice cores as a function of depth below the ice surface, and between the cores of the water bodies. For a simple comparison between ice cores and the three water bodies, mean values and the range (min., max. values) were calculated for every location (Tab. 2). The length of the ice cores, according to the thickness of the ice cover during field work ranged from 110 to 197 cm. Mean lengths were 144 cm in TB, 166 cm in PF, and 160 cm in GL. The ice of all cores from TB was nontransparent, with a high concentration of enclosed gas bubbles. The uppermost 3 cm (core 27) to 10 cm (core 29) was idenitfied as regelation ice from snow melt. The snow thickness ranged from 11 cm (core 27) to 32 cm (core 29). For PF, the ice of the two cores was clear and contained inclusions of elongated gas bubbles. The uppermost 3 cm of core 32 appeared milky-white. The difference in snow thickness above the two cores was quite large with 8 cm (cores 31) and 20 cm (cores 32). For GL, the ice morphology was heterogeneous. Core 20 appeared transparent-white, while core 21 included small, elongated

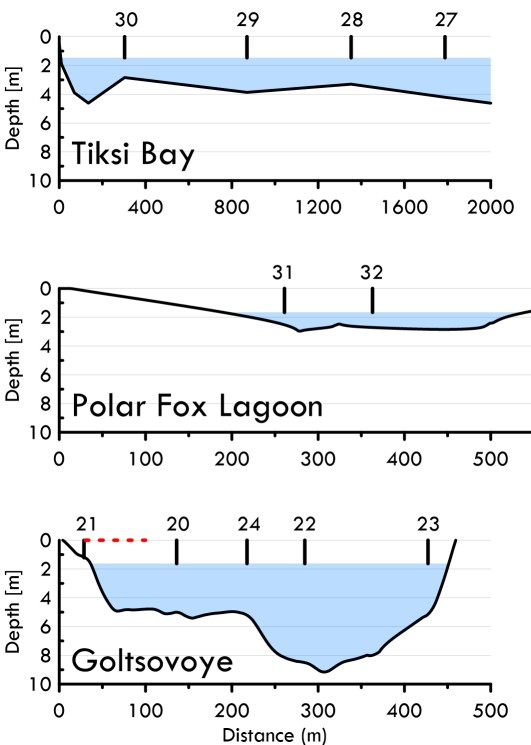

**Figure 2.** Cross-sections of the bathymetry of the Tiksi Bay (TB) profile, Polar Fox Lagoon (PF) from northwest to southeast) and Goltsovoye Lake (GL) along the coring transect. Approximate positions of the ice cores are indicated as numbered vertical lines. The position of the bubble transect at Goltsovoye Lake is represented with a dashed red line.

air bubbles and the uppermost $5\,\mathrm{cm}$ was regelation ice from snow melt. The ice of core 22 appeared whiter below a depth of $111\,\mathrm{cm}$ on, while core 23 was milky-white from the surface until the depth of about 112 to $114\,\mathrm{cm}$. The snow cover was generally solid and characterized by different melt forms. The thickness of the snow-layer ranged from $0\,\mathrm{cm}$ in snow free cores 20 and 24 to max. $92\,\mathrm{cm}$ in core 23. For all cores, no algae inclusions were visible. (Strauss et al., 2018).

## 4.2 Hydrochemistry of the ice

### 4.2.1 Electrical conductivity

All cores of TB had electrical conductivity values ranging from 340 to $2\,065\,\mathrm{\mu S\,cm^{-1}}$ (Tab. 2). All four cores showed a similar vertical trend: the electrical conductivity mainly increases with depth in the upper portion (until approximately 80-90 cm), and then decreases in the lower portion of the cores (>80-90 cm) (Fig. 3). For PF, the electrical conductivity ranged from 101 to $3\,630\,\mathrm{\mu S\,cm^{-1}}$ for both cores (Tab. 2). In both cores, electrical conductivity shows a similar increase with depth (Fig. 3). The electrical conductivity was lower in the upper portion of the ice and highest at the bottom. In GL, electrical conductivity is uniformly distributed with depth, and values ranged from $2.10\,\mathrm{\mu S\,cm^{-1}}$ to $150\,\mathrm{\mu S\,cm^{-1}}$. Electrical conductivity measured in



the ice of the freshwater GL is overall low, with a highest value of $150\,\mathrm{\mu S\,cm^{-1}}$ (core 23), compared to the marine influenced locations, TB and PF.

**Table 2.** The **mean** ±SD values and ranges (in parentheses) of measured parameters for all ice core samples for each site are shown.

| Parameter | Sampling Site | | | | | |
|---|---|---|---|---|---|---|
| | Tiksi Bay (**TB**) | | Polar Fox Lake (**PF**) | | Goltsovoye Lake (**GL**) | |
| Salinity | **0.64** ±0.26 | (0.16–1.05) | **1.02**±0.57 | (0.05–1.91) | **0.02**±0.02 | (0.00–0.07) |
| EC [$\mathrm{\mu S\,cm^{-1}}$] | **1276** ±498 | (340–2065) | **1979**±1075 | (101–3630) | **34.4**±45.6 | (2.10–150) |
| pH | **7.03** ±0.25 | (6.43–7.66) | **7.06**±0.28 | (6.63–7.69) | **6.15**±0.95 | (4.89–8.45) |
| Temperature [°C] | **-4.25** ±1.74 | (-7.25 – -1.05) | **-6.53**±3.90 | (-15.0 – -0.60) | **-2.68**±1.86 | (-7.30 – 0.20) |
| DOC [$\mathrm{mg\,L^{-1}}$] | **1.92** ±0.42 | (1.00–2.93) | **2.47**±0.70 | (0.66–3.55) | **0.86**±0.58 | (0.41–2.92) |
| CH$_4$ [nM] | **6.03** ±1.22 | (3.48 – 8.44) | **54.7**±109 | (2.59-539) | **645**±2282 | (0.02–14817) |
| $\delta^{13}\mathrm{C_{CH_4}}$  [‰] | **-46.0** ±3.76 | (-51.9 – -36.9) | **-51.0**±14.2 | (-79.7 – -31.8) | **-50.0**±15.9 | (-91.6 – -12.3) |
| $\delta^{18}\mathrm{O}$  [‰] | **-15.7** ±0.58 | (-16.7 – -14.7) | **-16.2**±0.64 | (-17.1 – -15.0) | **-17.8**±3.00 | (-28.4 – -16.3) |
| $\delta\mathrm{D}$ [‰] | **-120** ±4.33 | (-129 – -114) | **-123**±4.33 | (-130 – -114) | **-140**±21.0 | (-214 – -129) |
| d-excess | **5.0** ±0.5 | (3.5 – 5.7) | **6.0**±1.2 | (3.9 – 8.8) | **2.4**±3.3 | (-0.8 – 13.1) |

### 4.2.2   Temperature

Fig. 3 shows the average ice temperatures for each core at all 3 sites. Ice temperatures for TB ranged from $-7.3\,°\mathrm{C}$ at the
5   surface to $-1.1\,°\mathrm{C}$ at the bottom (Tab. 2). Overall, the temperature for all ice cores increases with depth (Fig. 3). For both cores in PF, the temperature ranges from $-15.1\,°\mathrm{C}$ (first 10 cm) to $-0.6\,°\mathrm{C}$ (bottom depths) (Tab. 2). The temperature of the four cores of GL was in the range from $-7.3$ to $0.20\,°\mathrm{C}$ (Tab. 2).

### 4.2.3   Dissolved organic carbon (DOC)

For TB, the DOC concentration ranges from $1.0$ to $2.9\,\mathrm{mg\,L^{-1}}$ (Tab. 2). The concentrations increase slightly with depth in
10   the upper portion of the cores. At the lower portion, the concentrations decreased, but at the bottom depths they increased again (Fig. 3). For the cores of PF, DOC concentrations increase with depth, with values up to $3.6\,\mathrm{mg\,L^{-1}}$ in the lower ice. In





the upper ice, DOC concentrations were highly variable and range from $0.7\,\mathrm{mg\,L^{-1}}$ to $3.3\,\mathrm{mg\,L^{-1}}$ (0-60 cm) (Fig. 3). DOC concentrations are between $0.7\,\mathrm{mg\,L^{-1}}$ and $3.6\,\mathrm{mg\,L^{-1}}$ for both cores, with a mean value of $2.5\pm0.7\,\mathrm{mg\,L^{-1}}$ (Tab. 2). The lowest DOC concentrations of the three locations were observed at GL. Here, the minimum concentration is $0.4\,\mathrm{mg\,L^{-1}}$ and the mean concentration $0.9\pm0.6\,\mathrm{mg\,L^{-1}}$ (Tab. 2).

### 4.2.4  Stable water isotopes

For the four cores of the TB transect, $\delta^{18}O$ and $\delta D$ values range from $-16.7\,\text{‰}$ to $-14.7\,\text{‰}$ and $-129\,\text{‰}$ to $-114\,\text{‰}$, respectively (Tab. 2). However, changes of the stable isotope composition with depth in the ice cores of TB are relatively similar to each other. While the values in the upper portion (until approximately 80-90 cm of the cores) are quite stable, except for the top of the ice, values decrease in the lower portion (below 80-90 cm) (Fig. 4). The values are lowest at the bottom of the ice cores in each case. The d-excess (d-excess = $\delta D$ - 8 $\delta^{18}O$ (Dansgaard, 1964)) is quite stable with a range from $3.5\,\text{‰}$ to $5.7\,\text{‰}$ for all four cores of TB, generally (Tab. 2).

For PF, the stable water isotope compositon ranges between $-17.1\,\text{‰}$ to $-15.0\,\text{‰}$ for $\delta^{18}O$ and $-130\,\text{‰}$ to $-114\,\text{‰}$ for $\delta D$ (Tab. 2). From the core surface to an intermediate depth of $60\,\mathrm{cm}$, isotope values increase, followed by decrease towards lower values at the bottom of the core. The d-excess values are stable from 10 to $20\,\mathrm{cm}$ until $90\,\mathrm{cm}$, followed by an increase near the bottom of the cores (Fig. 4.

For GL, the values of the stable water isotopes range from $-28.4$ to $-16.3\,\text{‰}$ for $\delta^{18}O$ and $\delta D$ values from $-215$ to $-129\,\text{‰}$ (Tab. 2). Both $\delta D$ and $\delta^{18}O$ of cores 20-24 do not vary with depth and are similar to the isotope signatures of TB and PF. Only core 23 displays significantly lower $\delta$-values (Fig. 4). Values for the d-excess were between $-0.8\,\text{‰}$ and $13.1\,\text{‰}$ (Tab. 2). For the cores 20, 21, 22 and 24, d-excess values range from $-0.8$ to $3.6\,\text{‰}$ (Fig. 4).

## 4.3  Methane in the ice

### 4.3.1  Dissolved methane concentrations

In the ice cores from TB, methane concentrations were generally very low (3.48-8.44 nM, mean: 6.03±1.22 nM) (Tab. 2) compared to the other locations and showed no variations with depth (Fig. 5). The dissolved methane concentrations in the water beneath the ice cores are $51\,\mathrm{nM}$ and $154\,\mathrm{nM}$ (core 29 and 30). In the ice cores from PF, we observed high methane concentrations at the upper part (up to 193.7 nM), followed by a decrease at depths from 60 to $100\,\mathrm{cm}$ and a final increase towards $539\,\mathrm{nM}$ at the bottom of core 32. In PF ice methane concentrations cover a wide range, with values from $2.59\,\mathrm{nM}$ to $539\,\mathrm{nM}$ (Tab. 2, Fig. 5). In the ice cores of PF, we observed high methane concentrations at the upper part (up to $194\,\mathrm{nM}$), followed by a decrease at depths from 60 to $100\,\mathrm{cm}$ and a final increase at the bottom of core 32. Concentrations of $1\,768\,\mathrm{nM}$ and $3\,394\,\mathrm{nM}$ (core 31 and 32) were measured for the dissolved methane concentrations in the water beneath the ice cores. In the ice cores of GL, methane concentrations are low and uniform with depth in the cores 20, 21 and 22. Very high and very variable concentrations (maximum $14\,817\,\mathrm{nM}$ in core 24) were observed in the cores 23 and 24, which both contained bubbles.



**Figure 3.** Vertical distribution of the electrical conductivity (EC), temperature and dissolved organic carbon (DOC) in ice cores with depth for TB (blue), PF (red) and GL (green).


**Figure 4.** Vertical distribution of the stable water isotopic compositions ($\delta^{18}$O, $\delta$D and d-excess) in ice cores with depth.



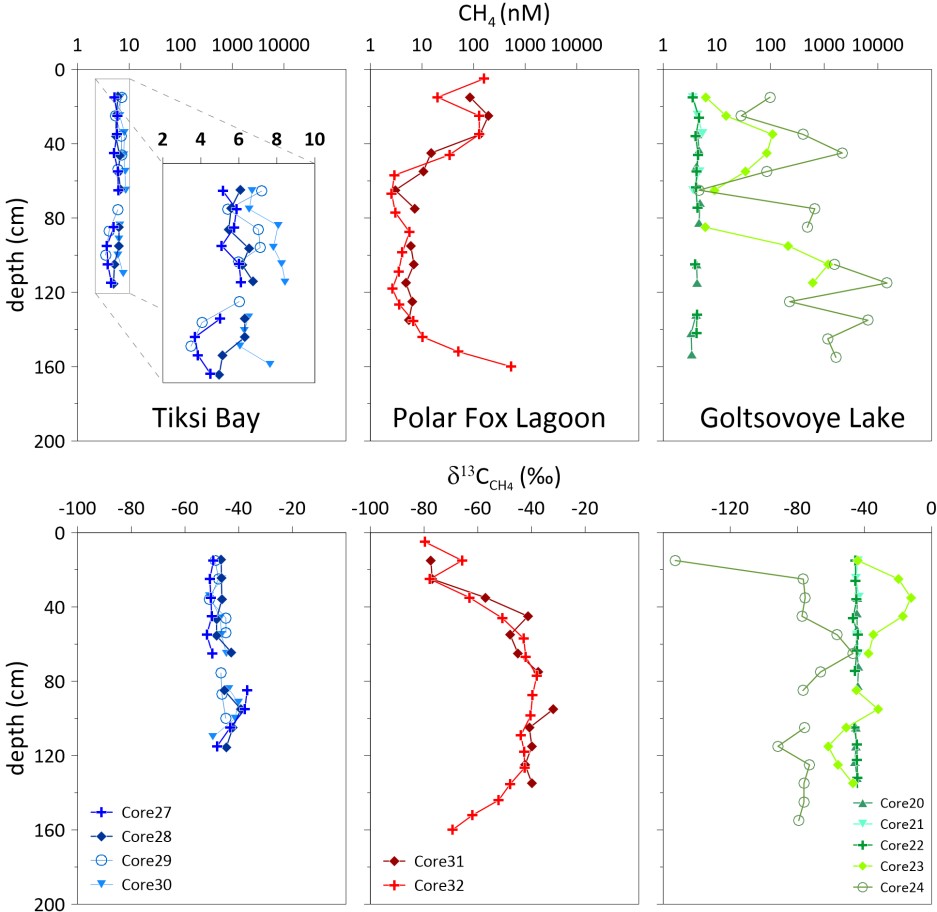

**Figure 5.** Vertical distribution of the methane concentration (on a logarithmic scale) and $\delta^{13}C_{CH_4}$ with depth for the three locations.

## 4.4 Spatial bubble distribution

In the total snow-cleared transect area of $89.2\,\mathrm{m^2}$ we found 29 seeps of class A and five seeps of class B (Fig.6), but none of larger classes C or hot spots (Walter Anthony and Anthony, 2013). The average density of class A seeps in the observed area is 0.33 seeps per square metre. Seep density of class B was more than 6 times lower (0.05 seeps per square metre). Total seep
5   density over the transect (classes A+B) was 0.39 seeps per square metre. The distribution of seep density along the transect shows no linear or homogeneous pattern.

### 4.4.1 Stable carbon isotopes

In the ice cores from TB, $\delta^{13}C_{CH_4}$ values ranged from $-51.9\,‰$ to $-36.9\,‰$. The values are in a smaller range than the $\delta^{13}C_{CH_4}$ values for the cores of the other locations (Tab. 2). In PF, the $\delta^{13}C_{CH_4}$ values range from $-79.7$ to $-31.8\,‰$ for both cores
10   (Tab. 2). The cores indicate a similar pattern, with carbon isotopes more enriched in $\delta^{13}C_{CH_4}$ in the lower portion of the cores

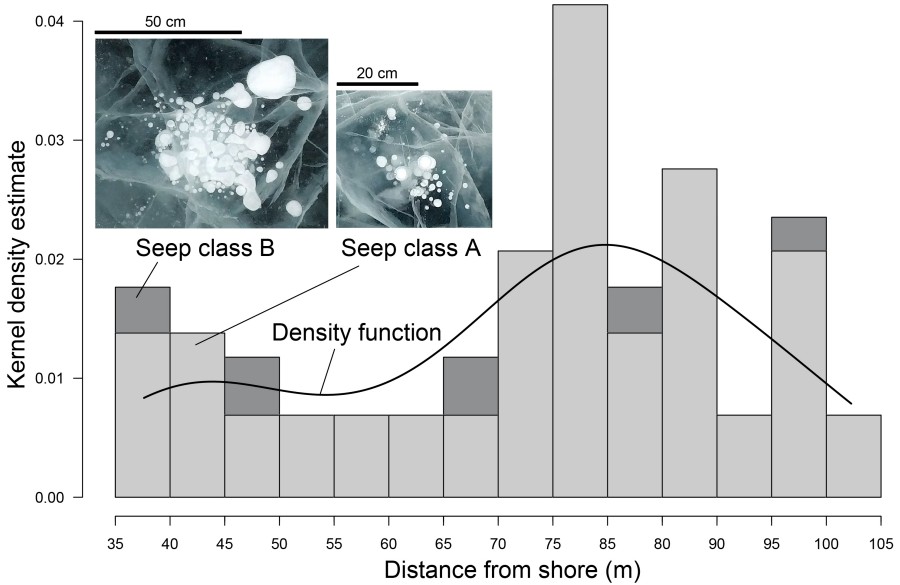

**Figure 6.** Distribution of methane seeps class A and B (Walter Anthony and Anthony, 2013) in Goltsovoye Lake perpendicular to the northwestern shore in an ice-freed area of 70 x 1.4 m. Bars represent the frequency of seeps A and B given as probability. The black line shows the Kernel density function (Gaussian Kernel performed on distances to shore, width is 34 data points).

than in the upper portion (Fig. 5). The stable $\delta^{13}C_{CH_4}$ values decrease in the bottom depths of core 32. The pattern is similar but inverse to the methane concentrations. In GL, the $\delta^{13}C_{CH_4}$ values range from $-91.6$ to $-12.3\,‰$ (Tab. 2). The highest and lowest values occurs at core 23 and 24. These two cores show changes in the $\delta^{13}C_{CH_4}$ values with depths, whereas the stable carbon isotopic signal of the other cores (20, 21, and 22) varies between $-46.8$ to $-43.3\,‰$ and is quite stable. In cores 20, 21 and 22, the $\delta^{13}C_{CH_4}$ values had a mean of $-43.3\,‰$ and were uniform ($\pm 2.2\,‰$) with depth, in contrast to cores 23 and 24, where values ranged from $-91.6$ to $-12.3\,‰$, with a strong variability within and between the two cores. Greater variability was observed for methane concentrations.

## 5   Discussion

A seasonal ice cover is a barrier to gas exchange between water and the atmosphere. The coastal water bodies studied here are covered by ice for 9 months of the year, a period that is shortening for both lake and marine ice (e. g. Günther et al., 2015). At the same time, air temperatures (Boike et al., 2013), sea level rise rates (Nerem et al., 2018), and coastal thermo-erosion rates (Günther et al., 2013) all increase, indicating a rapidly shifting regime for aquatic environments in the region. The importance of the persistence and duration of the seasonal ice cover in all of these processes is poorly understood. It may act not only as a barrier, but also as a source or sink for methane or as a habitat for microbes that facilitate methane consumption. As further warming of the Arctic shortens the duration of ice cover, pathways to methane emissions will probably shift. The three water



bodies in this study represent 1) a terrestrial thermokarst lake (GL), 2) a thermokarst lake that has become a coastal lagoon via thermo-erosion (PF) and the marine shoreface, in a setting where ice-rich Yedoma permafrost is undergoing thermo-erosion (TB). Although the ice cover sampled from all three settings was largely clear ice, important geomorphological differences between sites necessarily should lead to differences in methane dynamics and hydrochemical characteristics. In the lagoon

setting, thickening of the ice cover eventually plugged the shallow connection of the brackish basin to the sea, at which point freezing concentrated the remaining brine beneath the ice for the rest of the winter (**semi-closed system**), a typical situation in lagoons, behind barrier islands or on gently inclined shorefaces. Landfast ice on the shoreface does not close off the water basin, which continues to undergo exchange with the central Laptev Sea and Lena River inflow (**open system**). The thermokarst lake had neither outlets nor significant inlets, and the ice effectively closed off the water body (**closed system**), isolating the

freshwater basin and its talik. Based on our data, we suggest that the type of water body also determines the circulation of methane. Our results have consequences for the methane source-sink balance in polar terrestrial aquatic systems. In the following we examine each system's ice growth, methane concentrations and compositions and discuss the processes involved.

## 5.1 Tiksi Bay – the open system

Tiksi Bay is part of Buor Khaya Bay and via the central Laptev Sea perennially connected to the Arctic Ocean. The marine

impact delays the onset of ice formation compared to the terrestrial aquatic system shown by satellite images from ESA Sentinel-1 and -2. In addition to sea water, snowmelt in spring, small coastal catchments and the Lena River enter into TB between Cape Muostakh and Muostakh Island and around the southern end of Muostakh Island. Lena River discharge follows a nival discharge regime, with very high discharge in the spring and early summer months (Magritsky et al., 2018). While in winter when the connection between Buor Khaya Bay and TB is restricted by sea ice, the contribution of Lena discharge must

be much smaller. When TB freezes, it is supposed to be a continuously open system, i.e. water exchange is ongoing during winter below the ice. Both aspects (open system and the mixing of fresh and brackish water) are corroborated in the stable water isotopes composition of the ice cores. A mean $\delta^{18}O$ value of $-15\%o$ for TB is well below full marine conditions and displays the continuous and strong influence of freshwater supply through the Lena River to the Laptev Sea.

Firstly, in an open system such as TB, the water circulates freely beneath the ice cover, impeding the enrichment of lighter

water isotopes in the remaining water. Therefore, the isotope composition of the initial ice should remain more or less constant, and hence also that of the ice with depth (Gibson and Prowse, 1999), assuming the freezing velocity is roughly constant. Accordingly, the water isotopic composition and salinity values are stable until the ice is approximately 80-90 cm thick (Fig. 3,4). The small variation at the top might be due to variability in isotopic fractionation, i.e. related to a change of the freezing rate after ice formation started.

Secondly, TB is influenced by marine and river water, and a change in this ratio may change the isotopic composition of $\delta^{18}O$ and $\delta D$. The exchange of the water below the ice might have disturbed an equilibrium freezing process. Hence, water mixing is likely reflected in the small shift towards $-17\%o$ in $\delta^{18}O$ and $-130\%o$ in $\delta D$ in the lower part of the ice (>90 cm) (Figs. 3, 4, and 5), even though the $\delta^{18}O$-$\delta D$ data behave like isotopic fractionation in a closed system, where the last ice formed displays lightest isotope composition. The change of the water isotope composition without a change in the d-excess,



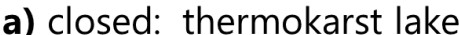

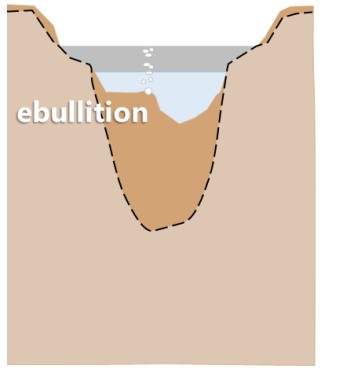

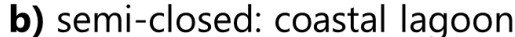

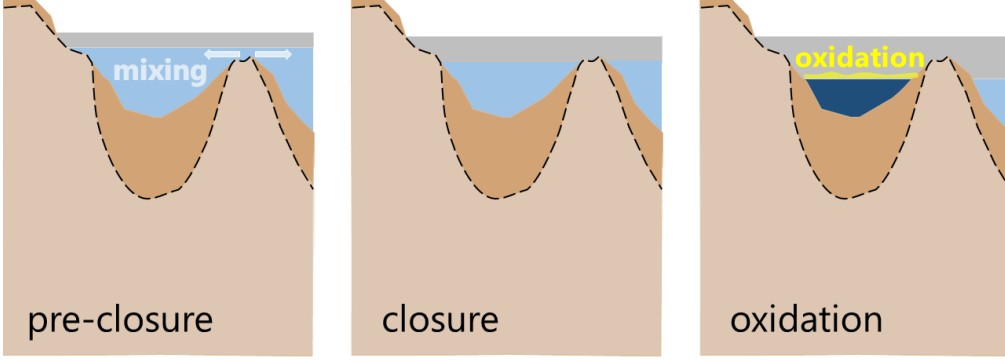

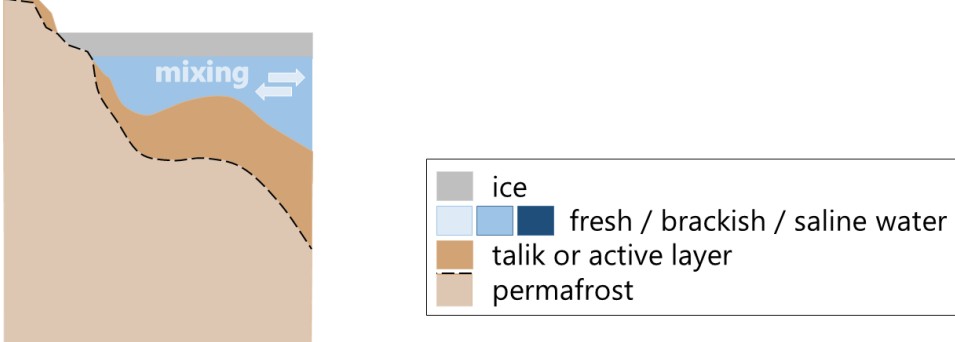

**Figure 7.** Schematics of geomorphological differences between the three aquatic systems studied in this paper: a) the closed system represents a typical thermokarst lake, such as GL, with a closed talik and a complete winter ice cover over freshwater; b) the semi-closed system is a brackish to highly saline water body (PF), whose ice cover separates it from the marine system at some point during the winter and for which continued ice growth results in hypersalinity; and c) the open system (TB), where ice cover growth does not prevent mixing with marine and or fluvial waters beneath the ice.





as well as changing EC and DOC, suggest a substantial contribution of another water mass and indicate the reduction of marine influence along or an increased river water impact in the lower part of the ice compared to the upper part. Reduced salinity and lower EC concentrations corroborated the enhanced freshwater content in the bottom segments of the ice cores, further substantiated by lower $\delta^{18}$O and $\delta$D values. Overall, the upper part of the TB ice cores indicate freezing from brackish water

whereas the lower part cmay apture a slightly higher proportion of river water probably reflecting changes to the isotopic and hydrochemical composition of Tiksi Bay. This is also shown in the regression lines of the $\delta^{18}$O-$\delta$D plot for TB, with obvious differences in the $\delta^{18}$O-$\delta$D slopes of the ice (Fig. 8), indicative for the preceding freezing process (Souchez and Jouzel, 1984). The samples of the upper portion of the ice plot along a regression slope of 6.9, demonstrating open system freezing under equilibrium conditions (Lacelle, 2011). In contrast, the lower portion of the ice plots along a regression line of 7.9 (Tab. **??**),

close to the slope of the GMWL of 8. Hence, we assumed this portion of the ice to be influenced by admixture of water of meteoric origin. The change in the water isotopes between the upper and lower ice is not related to closed system freezing, as this would also be indicated by a deviation from the GMWL and a change in the d-excess.

Methane concentrations in TB ice are small but still supersaturated relative to the atmospheric background concentration and probably the result of freezing of methane-supersaturated brackish water in TB. Supersaturation in the shallow water of TB is

probably caused by methane release from sediments (Bussmann et al., 2017), which can occur in the fall during resuspension events. Methane supersaturation has been reported for the Buor Khaya Bay (Bussmann, 2013) and large amounts of methane are stored in sediments of this region (Overduin et al., 2015). Lower methane concentrations during calm sub-ice winter conditions are indicated by lower methane concentrations in the lower part of the ice (>90 cm), but with the carbon isotope signature. The slight decrease in concentration with increasing distance from shore suggests calm water conditions during freezing and a local

source of methane. Very similar freezing processes for all TB cores are corroborated by the homogeneous ($\delta^{13}$C$_{CH_4}$) isotopic signature along the onshore-offshore gradient.

## 5.2 Polar Fox Lagoon – the semi–closed system

PF is connected to TB through a wide, shallow and winding channel. Storm surges and high water events in the summer force sea water, driftwood and sediment into PF. ESA Sentinel-1 and -2 imagery indicate that the ice on PF started to form in

mid-October, and hence earlier than at TB. During the early freezing period, the connecting channel allowed water exchange between PF and TB, until the growing ice reached the channel bed (less than 0.5 m deep) where it exits the lagoon in 2017. At that point, the PF lagoon changed from a connected to an isolated system, affecting water chemistry below the ice. The timing of closure, interpreted from the change of $\delta^{18}$O and $\delta$D values with depth, corresponds to an ice thickness of 60 cm.

Above this depth, the isotopic signature indicates freezing under equilibrium conditions (Lacelle, 2011)), with a slope of 8.2

between $\delta^{18}$O and $\delta$D (Fig. 8). Below this depth, however, isotopic values decrease, indicating freezing under closed conditions. Freezing leads to a lighter isotope signature of the remaining water, as the heavy isotopes are preferentially incorporated into the ice (slope: 6.5 Gibson and Prowse, 1999, 2002). This results in lighter isotope composition in later (deeper) ice and an increase of d-excess compared to earlier (upper) ice. The concentrations of dissolved constituents (indicated by EC and DOC) also increase with increasing ice thickness (Fig. 3), corroborating this interpretation.





The PF lagoon is on average around $1.5\,\mathrm{m}$ deep. At the time of coring the ice thickness was about $1.6\,\mathrm{m}$; consequently the outer lake area, more than $50\,\%$ of the lagoon area, was assumed to be frozen to the bed (Strauss et al., 2018). Thermokarst lake basins that are transformed into thermokarst lagoons may be increasingly affected by seawater, at least intermittently, during high water events such as storm surges, resulting in changes to their temperature and salinity regimes (Romanovskii

et al., 2000). Increasing salinity in turn also affects subsurface permafrost thaw dynamics and may therefore result in different methane production rates (Angelopoulos et al., 2019). In the uppermost ice, i.e. when freezing started, methane concentration in PF water was more than ten times higher than in TB. The isotopic signature was in the range of microbially produced methane ($-70$ to $-80\,\text{‰}$) (Whiticar, 1999). This large excess of $\delta^{13}$C-depleted methane clearly points to methane from the unfrozen talik, released into the water body and stored therein during the ice-free season.

When PF switched from a connected to a closed system at around $60\,\mathrm{cm}$ ice thickness, ongoing methane oxidation beneath the ice lowered the methane concentration captured in the ice to $\leq 10\,\mathrm{nM}$, comparable to those in TB ice. At the same time, the methane isotopic signature became comparably enriched in $\delta^{13}$C (Fig. 5, 9). Ongoing ice formation under closed system conditions (below $60\,\mathrm{cm}$), as indicated by stable water isotopes, induced a continuous increase in ice salinity (Fig. 3) which in turn favoured the shift of the horizon where methane oxidation could occur from the water to the bottom of the ice. This is

corroborated by the Rayleigh fractionation curves calculated for ice that grew under closed conditions, using as initial methane isotopic signature the uppermost value, i.e. the signature when freezing began (Fig. 5, 9).

In addition, temperature increases towards the bottom of the ice (Fig. 3). The bottom ice offers a protected environment with favourable conditions for microbial metabolism: relatively warm and stable temperatures, contact with liquid water and permeable ice, permitting migration of gases and nutrients, similar to marine ice, where most bacteria are located in the lowest

centimetres of the ice (Krembs and Engel, 2001). During freezing of the ice cover, its growth rate decreases (cf. Anderson, 1961), providing more time and space for bacterial metabolism. Methane uptake from the water into the bottom of the ice and its oxidation there may have continued over the winter until the ice break-up. Methane oxidation ceases when concentrations are too low for oxidation to be efficient (Cowen et al., 2002; Valentine et al., 2001), at values ranging from $0.6\,\mathrm{nM}$ to $10\,\mathrm{nM}$. Methane concentrations in the ice above $130\,\mathrm{cm}$ (Fig. 5) are less than $10\,\mathrm{nM}$, suggesting that ice is an effective sink for methane

in this type of water body during winter.

## 5.3   Goltsovoye Lake – the closed system

The ice core hydrochemistry from Goltsovoye reflects a water body that freezes in euqilibrium with atmospheric methane concentrations with two cores showing the influence of snow loading and of ebullition.

Goltsovoye Lake (GL) is an isolated thermokarst basin surrounded by ice-rich Yedoma uplands to the west and east and

partially degraded Yedoma uplands to the north and south. The lake is underlain by continuous permafrost some hundreds of meters deep, and has a thaw bulb (talik) beneath its bed due to the positive temperatures at the lake bottom. Water in GL derives from precipitation, most of which falls as snow, overland flow and perhaps as groundwater flow through the shallow active layer. Thus, the concentration of dissolved constituents remains small in lake water and ice, as reflected in very low electrical conductivities of less than $50\,\mu\mathrm{S\,cm^{-1}}$ (Fig. 3).





Water depths range from < 1m to about 8.5 m from the western to eastern shore, respectively (Fig. 2). This asymmetric shape influences the progress of ice growth in winter. Ice formation typically starts from the lake shore, most likely at the very shallow west shore, and leads to bedfast ice formation at the position of core 21, i. e. lake ice frozen to the lake bed. Hence, it is likely that core 21 began to form earlier in the season, and that the upper ice in this core, and probably also in core 22,

reflects the chemistry of the summer/autumn lake water. All cores except core 23 had similar $\delta^{18}O$ (around -16.5‰) and $\delta D$ (-140‰) values, and with a regression slope of 6.6 (Fig. 8), pointing to freezing under equilibrium conditions (Lacelle, 2011). At this location both, the methane concentrations and the uniform $\delta^{13}C$ signature indicate equilibrated values with respect to the atmospheric background values (Fig. 5). This circumstance clearly shows that lake water was not supersaturated during the ice-free season. The other two cores show clear differences and are described in the following.

For core 23 stable isotopes in the upper 120 cm follow the GMWL (Fig. 8), indicating equilibrium fractionation (in the system vapour-water, as snow is involved) due to different precipitation sources. $\delta^{18}O$ values as low as -28‰ indicate the involvement of snow. The proximity of snow samples to the GMWL is typical for Northern Siberia and has been also found at snow patches on the Bykovsky Peninsula (Meyer et al., 2002). Core 23 was taken proximal to the lake's steepest shoreline, where active thermo-erosion results in shoreline retreat, and that lies in the lee of prevailing winds (Günther et al., 2015),

where deep snow is expected to accumulate and load the lake ice. 92 cm thick snow lay on the ice at this location at the time of drilling and water streamed out of the hole after coring (Strauss et al., 2018), indicating an ice cover under positive hydrostatic pressure. The ice of core 23 was milky-white from the surface to about 112 to 114 cm depth consistent with a mixture of snow and water. Thus, we conclude a snow signal evident from the stable isotope composition in the upper 120 cm of core 23. The higher EC and DOC in the same interval (relative to lower ice and to cores 20-22 & 24) are unlikely to have derived from

snow, however, and imply heterogeneous ice development above 120 cm. These may be the result of mixing of uprising lake water with snow within ice cracks. Adams and Lasenby (1985) describe the formation of white ice (or snow ice) by water percolation through thermally-induced cracks to the surface of the ice, where the water mixes with snow and forms another ice layer above the former ice. This was observed by Adams and Lasenby (1985) when a snow load depressed the surface of the ice cover below the hydrostatic water level. The high and highly variable methane concentrations over this interval, together

with the high EC and DOC, suggest that resuspension events, for example slope failure, occurred during ice formation. The carbon stable isotope signature in core 23 may reflect microbial methane oxidation in the sediment rather than oxidation in or beneath the ice.

The highest concentrations of methane were found in core 24 (up to 15 000 nM), with the lowest $\delta^{13}C$ signatures (down to $-150$‰, Fi. 5). Core 24 lwas drilled above the steepest portion of the lake bed, where ebullition may be a by product of

thermokarst processes beneath the bed. The $^{12}C$-enriched signal is consistent with methane that has not been oxidized in the sediment and is released in gas bubbles. The earlier release of the lightest methane may be the result of fractionation during migration in the sediment and may be an indication the ebullition and ice formation are related, i.e. that ebullition was initiated as a result of pressure changes owing to the onset of freezing.



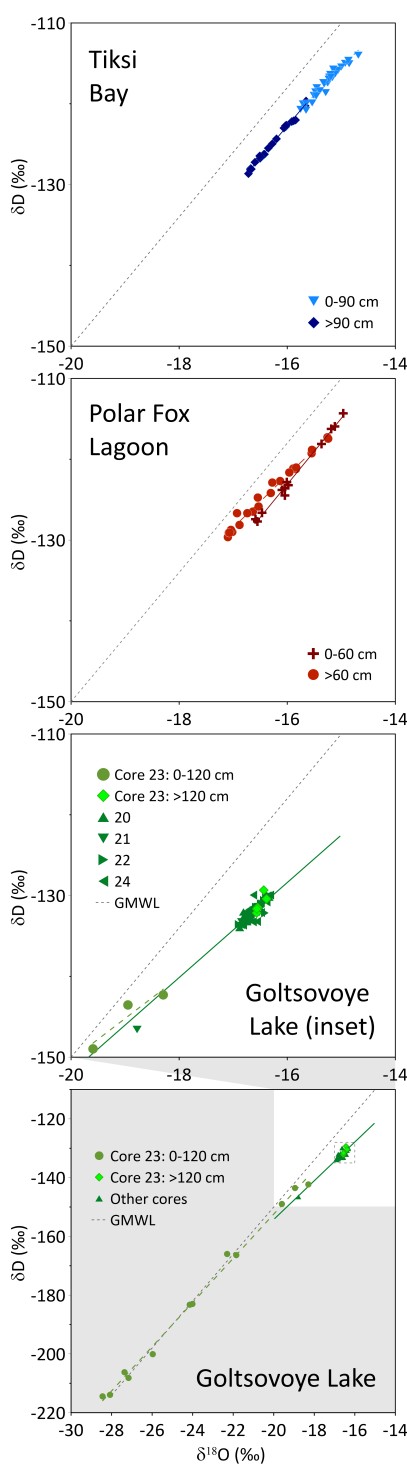

**Figure 8.** Isotope composition and linear regressions for TB (top), PF (middle) and GL (bottom) ice cores. For comparison, the global meteoric water line (GMWL, $\delta D = 8 * \delta^{18}O + 10$; (Craig, 1961), grey dotted line) is shown.



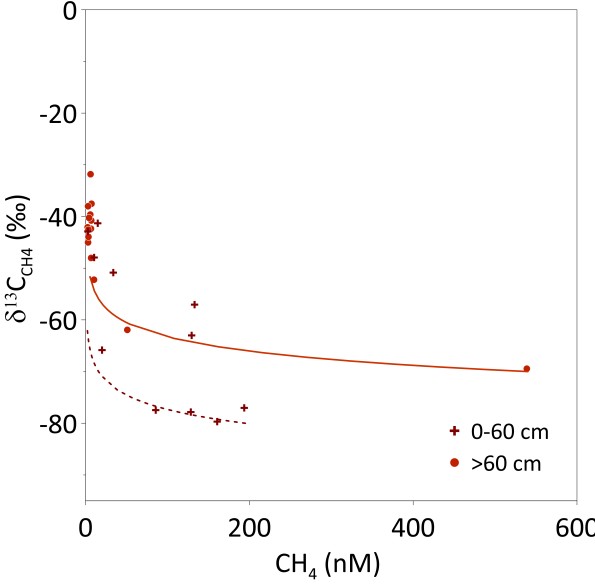

**Figure 9.** Methane oxidation in the ice core of Polar Fox calculated by a Rayleigh fractionation model (see text). The curves show the prospective $\delta^{13}C_{CH_4}$ signatures modified by oxidation (lines). The methane consumed by oxidation was calculated by $\alpha = 1.004$. The assumed initial concentrations were taken from surface ice: (195 nM (PF open, red dotted line) and 450 nM (PF closed, red solid line). The initial isotopic signatures were $-80‰$ (PF open, red dotted line) and $-70‰$ (PF closed, red solid line).

## 6 Conclusions

Methane concentrations in the seasonal ice cover of three types of Arctic water bodies, representing three different stages of permafrost degradation, revealed differences related to the process of ice formation and its importance as mitigator of methane pathways.

5    In the ice of Tiksi Bay, which is open to the central Laptev Sea throughout the winter and also underlain by permafrost, the stable isotope signatures and electrical conductivity suggest overall brackish conditions, with an increased admixture of riverine waters with ongoing freezing. Ice composition reflected the composition of the upper layer of brackish water throughout the winter and methane concentrations were low but supersaturated.

   In the coastal Polar Fox Lagoon, connected to the sea during summer, ice formation and sealing of a connecting channel 10  between lagoon and Tiksi Bay closes off the water body during freezing, isolating and concentrating remaining brackish water beneath the thickening ice during the winter. In the earlier stages of freezing, the lagoon is still connected to Tiksi Bay and shows similar, brackish conditions in stable isotopes and electrical conductivity. In the later stages of freezing, the lagoon is separated from the bay's influence and behaves like a closed system with decreasing $\delta^{18}O$ and $\delta D$ and increasing d excess values. Methane is present at variable concentrations in the lagoon, but the concentration profile over depth and the stable



isotope signatures suggest that bacterial oxidation takes place at the interface between ice and lagoon water, reducing the methane concentration preserved in the ice.

In a land-locked thermokarst lake surrounded by Yedoma landscapes, rather uniform $\delta^{18}O$ and $\delta D$ values and very low electrical conductivity in all lake ice cores (except for one) indicate either subsurface contributions to the lake in winter or a

lake deep enough not to behave like a closed system. The exceptional core had a clearly meteoric (likely due to snow-loading of the ice) contribution. Methane concentrations in the lake ice were spatially highly variable. High methane concentrations were local and probably associated with ebullition and snow loading of the ice at an eroding shoreline.

Overall, ice on coastal waters in this environment acts primarily as a barrier to methane fluxes to the atmosphere, a barrier that is effective for most of the year but also will be effected by rapid changes due to Arctic warming and associated ice

thinning. Additionally, we have shown that the ice cover may act as a sink, providing a habitat for methane oxidation.

As the sediment is a known environment for methane production and DOC could be a source for methane production in the water or the ice, sediment pore water $\delta^{13}C_{CH_4}$ values, and methane and DOC concentrations should be included in future studies to understand methane pathways from their source in comparable water bodies. Furthermore, the comparison between brackish and freshwater water bodies may yield insights into the constraints on methane oxidation in thermokarst lakes and

Arctic lagoons. As carbon dioxide is an important greenhouse gas and the product of methane oxidation, future studies should include relative proportions of both greenhouse gases.

*Data availability.*  Concentration data are available on PANGAEA (insert doi here).

*Author Contributions* IS, PPO, ED and IB conceptualized the study. Field investigations and/or laboratory analyses were carried out by IB, PPO, HM, SL, MA, BKB, MNG and GG. an IS prepared the manuscript with contributions from all co-

authors. The authors declare that they have no conflict of interest.

*Acknowledgements.*  This study was funded in part by the project PETA-CARB (ERC #338335). The Russian Foundation for Basic research supported this work via projects awarded to Mikhail N. Grigoriev (18-05-70091, 18-45-140057). Logistical support for the fieldwork was provided by the Russian Hydrographic Service (HydroBase Tiksi). We thank Hoang Vu Tuyen for rectifying and mosaicking images for the methane bubble transect and Ilsetraut Stölting for help with methane measurements.



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
