# Peer review of "Methane Pathways in Winter Ice of a Thermokarst Lake-Lagoon-Coastal Water Transect in North Siberia"

_The Cryosphere, 2019_

## Referee Comment (RC1) · Pat Langhorne (Referee) · 1 Jun 2020

This paper describes ice on three distinct water bodies, in particular examining the methane within the ice and the physical properties associated with understanding the observed methane concentrations. I am expert in the growth of ice on water bodies, but not on the suite of chemical techniques involved in the study and my comments below need to be read with this in mind. I believe that the study is interesting and deserves publication but could be made easier for the reader to understand, especially those not already expert in every aspect of the work.

Comment 1: The paper deals with three distinct water bodies which are expected to

show different characteristics. Yet there is nothing in their abbreviated names that helps the reader immediately recognize which water body is being described. Why not call the sites Bay, Lake, Lagoon, or some abbreviation that is easily recognizable, such as BY, LK, LG?

Comment 2: In addition a number of cores are taken at each site, and interesting behavior is shown in these data in Figures 3-5. However the naming of the cores could better reflect their position and make it easier for the reader to interpret this behavior– for example the TB cores could be labeled from N to S, while GL cores from E to W.

Comment 3: In my opinion there was too much description of the shape of graphs etc at the expense of what the reader might expect to learn from that particular type of behavior. I understand that the paper is the work of a thesis, but while such description is appropriate for a thesis it extends the length of a journal article unnecessarily. For example, sections 4.2.2, 4.2.3, 4.2.4, 4.3.1 and 4.4.1 give detailed description of what can be seen by looking at Figs 3, 4 and 5, and Table 2. What I wanted to know was what can be scientifically deduced from the observed values of parameters, or form of graphs. I suggest that the authors replace these detailed descriptions with the scientific evidence provided by the particular behaviors.

Comment 4: I felt that the authors tended to make rather grandiose statements that were not obviously dealt with in the paper e.g. statements regarding ice as a barrier to methane fluxes and the importance to warming in the Arctic. I suggest that authors carefully consider what can be deduced from their work and focus on the aims of their study. Comment 5: I do not have the expertise to critically review the chemical techniques used in this paper and whether they are appropriate and carefully carried out. I'm afraid that the editors must seek advice another reviewer for that expertise.

Technical Corrections

p. 1: Abstract: Please rewrite, taking into consideration the Comments above.

[Figure]

p.1, line 3: ".. provide insights on methane pathways in winter ice cover.." But at the end of the paper I had not recognized what these insights were, nor could I find it in the Abstract.

p.1, line 10: "except for three"

p. 1, Line 14: Comment that "methane oxidation may decrease methane concentrations during winter" Where? In the ice? In the atmosphere? Both? Is this the evidence for the winter pathway but I have not recognized it?

p. 1, Line 14-16: I could not follow how methane pathways in freshwater systems led to the understanding of permafrost carbon feedbacks in global warming – this seemed to be a huge leap to me – but perhaps I show my lack of knowledge of permafrost.

p. 2, Line 27-28: This is not a sentence

p. 3, Line 20-23: Here the authors clearly outline the three aims of the paper. I am clear that they have achieved the first aim, but I am unclear regarding aims 2 and 3. I return to this comment in the Conclusions.

p. 3, Line 20-23: "The Bykovsky.."

p. 4, Line 3: please give approximate depths

p. 5, Table 1: The Table implies that PF had a temperature constant to 0.01 oC over its 4 m depth. I found this unlikely. Please justify.

p. 5, Lines 10-16: I don't see the point in telling us about data collected that is not analysed in the article.

p. 5, Lines 15: What does "res" mean?

p. 6, Sect 3.2: I think information about transport and storage of the cores is missing (e.g. temperature) and seems as important as other details that are provided.

p. 6, Line 11: Define EC first time used.

p. 6, Line 12: "as soon as possible" is not very specific

p. 7, Line 2: "Slope"

p. 7, Line 23: "100 ppm that were"

p. 7, Line 27: "from the same bottle"

p. 8, Line 14: What is the "ice-free transect area" and why was it needed as scale in the photos?

p. 8, Line 22: What are "the cores of the water bodies"?

p. 8, Line 27: "identified"

p. 8, Line 27: How is regelation ice from snow melt identified?

p. 8, Sect 4.1: Difficult to follow as the reader needs to keep referring to which core numbers are from which site. A sketch of the ice types would reduce the need for detailed description and give a better overall view of the structure of the ice covers.

p. 9, Figure 2: Why does PF look so small on this figure? On Figure 1 no dimension of PH seems to be less than GL? In addition it would be good to label geographic location on the transects, i.e. N, S, NW etc

p. 9, Line 2: remove "on"

p. 10, section 4.2.4 & p. 15, 4.4.1: I suggest replacing "stable" with "constant". "Stable" implies "firmly fixed" and I am not sure that this is what you wish to imply. If you do mean "stable" then I think you need to justify why you expect no change under any change in circumstances.

p. 10, Line 12: spelling "composition"

p. 12, Fig 3: It is very interesting that PF is colder than GL. Is this because it is shallower?
p. 15, Fig 6 caption: "free". Has the ice free area been marked on Fig 1?

p. 15, Line 13-14: Is this statement tested in the present article, or is this speculation to introduce the Discussion?

p. 16, Line 10-11: It was not at all obvious to me how the data presented showed that the type of water body determines the circulation of methane. Please explain.

p. 16, Line 15: Suggest replacing "impact" with "setting"

p. 16, Line 18: "…(2018), while in winter, when the connection.."

p. 16, Line 26: Why would the freezing velocity be approximately constant? It is likely to decrease as 1/(ice thickness).

p. 16, Line 24-29: The authors note that they have not taken into account the freeze fractionation influences (e.g. see Toyota et al., 2013), based on the assumption that the freezing velocity is roughly constant. If this is not necessary please provide an order of magnitude calculation to convince the reader that this is a small effect.

Toyota, T., I.J. Smith, A.J. Gough, P.J. Langhorne, G.H. Leonard, R.J. Van Hale, A.R. Mahoney, and T.G. Haskell. (2013) Oxygen-isotope fractionation during the freezing of seawater. Journal of Glaciology. Vol. 59, No. 216, 2013 doi:10.3189/2013JoG12J163

p. 17, Fig 7: Very nice helpful sketches. They might be useful earlier in the manuscript.

p. 18, Line 2: "alone"

p. 18, Line 5: "may capture"

p. 18, Line 5-7: Interesting observation that may be compared with the results of Smith et al., (2016).

Smith, I.J., Eicken, H., Mahoney, A.R., Van Hale, R., Gough, A.J., Fukamachi, Y., Jones, J. (2016). Surface water mass composition changes captured by cores of Arctic land-fast sea ice. Continental Shelf Research, 118:154-164, doi:

10.1016/j.csr.2016.02.008.

p. 18, Line 7: "indicative for the preceding freezing process (Souchez and Jouzel, 1984)." I'm not sure what this means? Does this mean that freeze fractionation should be taken into account?

p. 18, Line 9: "Tab ??". This is not at all obvious from Fig 8.

p. 18, Line 18: "but with the carbon isotope signature". I could not see much change in the carbon isotope?

p. 18, Line 29: "(Lacelle, 2011)"

p. 18, Line 29: Again, this is not at all obvious from Fig 8.

p. 20, Line 29: "was"

p. 21, Fig 8: The changes in slope appear to be important but cannot be seen on the plots as currently displayed. Please consider how to display this information to match the Discussion. Why is global rather than local meteoric line used?

p. 22, Fig 9: Is it not possible to write down the equation of the model displayed?

p. 22, Conclusions: Please return to the aims of the study here, and show how they have been moved forward.

p. 23, Line 8: It is not obvious to me how the data provided has shown that the ice examined "acts primarily as a barrier to methane fluxes to the atmosphere, a barrier that is effective for most of the year but also will be effected by rapid changes due to Arctic warming and associated ice thinning."
 Please make this clearer in the Discussion and/or Conclusion.

p. 23, Line 8: What does "providing a habitat for methane oxidation
" mean? Again this needs to have been explained earlier in the paper.

---

## Referee Comment (RC2) · Blaize Denfeld (Referee) · 13 Jul 2020

This study attempts to improve our understanding of CH4 pathways in ice covered water bodies by focusing on ice and inland water continuum from lake to coast, both seldom included in the limited winter CH4 inland water studies. In doing so, the study highlights varying CH4 concentrations explained by the geomorphological differences between the aquatic systems. This is an interesting study with a unique data set and with revisions could be a nice addition to the scientific community.

General Comments:

[Figure]

1) The studies focus is on Arctic waterbodies underlain by permafrost. However, the introduction covers a broader range, i.e. when referring to Arctic and Northern lakes not all are in continuous permafrost zones. I think this broad perspective of northern lakes is appropriate but think the introduction could be better structured to go from a broad approach (Northern ice-covered lakes) to lakes in a continuous permafrost landscape. In particulate Paragraph 2 of the introduction would benefit from this restructuring.

2) Additional details needed to be clarified in the methods, particularly on CH4 lab sampling. See specific comments below.

3) The discussion is sorted into the three different water bodies, which works but it would be helpful to also have an overview of how the values calculated for these water bodies compare with values in the same system (lakes/coastal permafrost areas). Perhaps reporting a range for all 11 ice cores and discussing how it compares to other ice-covered inland water values. This could be done in the initial discussion paragraph before diving into the specifics of the three water body types.

4) In the conclusion, in addition to returning to the aims of the study it would be nice to know how these findings fit into understanding the lake-lagoon-coast transition in the arctic region. Although I appreciate that caution should be taken in making large upscaling statements given the limited sample size and snapshot in time.

5) Much of cited literature focuses on findings from lakes. Are there any studies that have looked at CH4 concentrations below ice in coastal areas? If so, it would strength the paper to include them in the introduction and discussion (see general point 3).

Specific Comments:

Title: Since the study only investigates one lake, lagoon and bay perhaps the title would better represent the actually study as, "Methane Pathways in Winter Ice of Thermokarst Lake-Lagoon-Coastal Water Transect in North Siberia"

P1, L 7-9: Could use a tie in sentence and possible move this information after L 5-6 as

it is continuing to point out differences between the system, e.g "In addition the three water bodies had different freezing systems. In TB. . ."

P1, L 12: is "above the ice-water interface" referring to in the ice? If so please clarify.

P 2, L 7-9: This sentence should be rewritten. Is the idea that CH4 can continue to accumulate in lakes over the ice-covered period whereas in soils the active layer freezes during winter and CH4 is not produced? A reference showing that CH4 production in the active layer of permafrost is mainly during summer would help support this statement. CH4 has been found to accumulate in shallow lakes over winter, so the authors may need to think about the definition of "certain circumstances".

P2, L 23-25: I assume these two sentences are referring to lake sediments? Please clarify.

P2, L 25: a third pathway, plant mediation, should be included.

P3, L 1-4: Are these the only two studies looking at CO2 and CH4 in ice? Perhaps these sentences could be simplified as one, "Of the limited studies, accumulation of CH4 in and under the ice during winter were realized for shallow ice-covered lakes in Alaska (Phelps et al. 1998) and four lakes in discontinuous permafrost area (Boereboom et al. 2012)." Or something like that.

Pg 3, L6, "However, methane oxidation. . ."

P3, L10: change to, "methane has been found to oxidize at temperatures as low as "C." Material and Methods:

P5, L 15: typo "res"

P6, L 1-9: How were the ice samples stored, in -15C? Could there have been CH4 oxidation? How effective was the vacuum pump at removing O2?

Pg 6, L 22-25: How many samples were considered high salinity and low salinity?

Pg 7, L 14: Remove Global Meteoric Water Line, already abbreviated above.

Pg 7, L 19-20: More details needed here. When the N2 was added did it create an overpressure in the vial or was 5 mL of water removed? How was the water sample equilibrated with the N2, shaken? Was the equilibrated air then removed from the vial and injected into the GC?

Pg 8, 10-19: Was the bubble transect done before or after the ice coring, i.e. were the ice core samples taken from the targeted plot area or were they randomly selected?

P 18, L 9: typo, "(Tab. ??)"

P 18, L 10: typo, "admixture"

P 19, L 12-16: Could it also be because less is being produced?

P 20, L 29: typo, "lwas"

Table 1: In the legend add "Water" at the start of the second sentence. For the electrical conductivity column could you report Salinity [PSU] /EC [mS/cm] since salinity is known for all three water bodies. Or have two sperate columns for salinity and EC.

Table 2: For PF replace Lake with Lagoon in the Sampling Site header.

Figure 4: GL has different scales for the first two rows

Technical Comments:

1) There are many places where Methane is written out, since it is abbreviated to CH4 on P1 L19 it should be changed to CH4 thereafter.

2) Keywords: remove . . . at end of list

---

## Author Comment (AC1) · 7 Sep 2020

Pat Langhorne (Referee)
pat.langhorne@otago.ac.nz

This paper describes ice on three distinct water bodies, in particular examining the methane within the ice and the physical properties associated with understanding the observed methane concentrations. I am expert in the growth of ice on water bodies, but not on the suite of chemical techniques involved in the study and my comments below need to be read with this in mind. I believe that the study is interesting and deserves publication but could be made easier for the reader to understand, especially those not already expert in every aspect of the work.

Comment 1: The paper deals with three distinct water bodies which are expected to show different characteristics. Yet there is nothing in their abbreviated names that helps the reader immediately recognize which water body is being described. Why not call the sites Bay, Lake, Lagoon, or some abbreviation that is easily recognizable, such as BY, LK, LG?

> Response: We changed the abbreviated names to Tiksi Bay (BY), Polar Fox Lagoon (LG) and Goltsovoye Lake (LK) throughout the paper. We have also changed the names and numbers of the cores to reflect the water body and sequential geographic position of each core.

Comment 2: In addition a number of cores are taken at each site, and interesting behavior is shown in these data in Figures 3-5. However the naming of the cores could better reflect their position and make it easier for the reader to interpret this behavior– for example the TB cores could be labeled from N to S, while GL cores from E to W.

> Response: we have relabeled the cores but used the abbreviations suggested in the previous comment (e.g. BY-1, BY-2, etc.) always in a uniform manner following compass direction. This has been applied consistently to all figures, the text, online data, and tables. We added the compass directions to the bathymetric profiles as well, to help the reader assign order to direction.

Comment 3: In my opinion there was too much description of the shape of graphs etc at the expense of what the reader might expect to learn from that particular type of behavior in the graph. I understand that the paper is the work of a thesis, but while such description is appropriate for a thesis it extends the length of a journal article unnecessarily. For example, sections 4.2.2, 4.2.3, 4.2.4, 4.3.1 and 4.4.1 give detailed description of what can be seen by looking at Figs 3, 4 and 5, and Table 2. What I wanted to know was what can be scientifically deduced from the observed values of parameters, or form of graphs. I suggest that the authors replace these detailed descriptions with the scientific evidence provided by the particular behaviors.

> Response: We have tried to be strict in separating the reporting of results from their interpretation. The interpretation of the results is therefore only found in the discussion section. We take this comment to mean that the results section is unnecessarily long and we have made numerous edits to shorten the sections mentioned (please see track changes section).

Comment 4: I felt that the authors tended to make rather grandiose statements that were not obviously dealt with in the paper e.g. statements regarding ice as a barrier to methane fluxes and the importance to warming in the Arctic. I suggest that authors carefully consider what can be deduced from their work and focus on the aims of their study.

We agree that there is definitely room for us to hone our objectives, discussion and conclusions in order to be more precise about what we did, why, and what its implications were. Please see our answers below regarding the objectives, discussion and conclusions, where we detail the deletions and editing we have undertaken. We feel the paper has been slimmed down and become more precise in its aim and conclusions.

Comment 5: I do not have the expertise to critically review the chemical techniques used in this paper and whether they are appropriate and carefully carried out. I'm afraid that the editors must seek advice another reviewer for that expertise.

Technical Corrections
p. 1: Abstract: Please rewrite, taking into consideration the Comments above.
    Adopted, see track changes version.
p.1, line 3: ".. provide insights on methane pathways in winter ice cover.." But at the end of the paper I had not recognized what these insights were, nor could I find it in the Abstract.
    We agree that this was imprecisely worded. We have replaced the 3rd and 4th sentences of the abstract:
    "The fate of methane in these waters and is poorly understood. We provide insights into the methane pathways in the winter ice cover on three different water bodies in a continuous permafrost region in Siberia."
    with:
    "How methane concentrations and fluxes in these waters are affected by the presence of an ice cover is poorly understood. To relate water body morphology, ice formation, and methane, we studied the ice of three different water bodies in locations typical of the transition of permafrost from land to ocean in a continuous permafrost coastal region in Siberia."
p.1, line 10: "except for three"
    Corrected.
p. 1, Line 14: Comment that "methane oxidation may decrease methane concentrations during winter" Where? In the ice? In the atmosphere? Both? Is this the evidence for the winter pathway but I have not recognized it?
    Response: the sentence has been re-worded and we have added "...on the lower ice surface."
p. 1, Line 14-16: I could not follow how methane pathways in freshwater systems led to the understanding of permafrost carbon feedbacks in global warming – this seemed to be a huge leap to me – but perhaps I show my lack of knowledge of permafrost.
    This sentence was deleted.
p. 2, Line 27-28: This is not a sentence
    Corrected.
p. 3, Line 20-23: Here the authors clearly outline the three aims of the paper. I am clear that they have achieved the first aim, but I am unclear regarding aims 2 and 3. I return to this comment in the Conclusions.
    Response: we agree, and have made the following changes:

    P 3 lines 18—22: we have changed the **objectives** of the study to be more precise and focussed, from
    "This study aims to clarify the role of an winter ice cover for methane cycles of three different stages in the lake-lagoon-shelf transition in a region of rapidly thawing permafrost in northeast Siberia. Our objective is to demonstrate how methane is distributed within seasonal ice from Tiksi bay (TB), Polar Fox Lagoon (PF), and Goltsovoye Lake (GL), to better understand 1) how freezing processes differ between the three water bodies, 2) what the relationships between freezing dynamics and methane concentration in the ice are, and 3) the potential importance of methane oxidation in different water bodies."

to:

"To improve our understanding of how water bodies function as CH4sources or sinks, this study aims to clarify the role of the winter ice cover for CH4 in three different stages in the lake-lagoon-shelf transition in a region of thawing permafrost in10northeast Siberia. Our objective is to demonstrate how CH4 is distributed within seasonal ice from Tiksi Bay (BY), Polar Fox Lagoon (LG), and Goltsovoye Lake (LK), to better understand 1) how freezing processes differ between the three water bodies, 2) how freezing affects CH4 concentration in the ice, and 3) to gain an indication of which processes change CH4 concentrationduring the ice cover season."

p. 3, Line 20-23: "The Bykovsky.."

Corrected.

p. 4, Line 3: please give approximate depths

Added depth range in parentheses.

p. 5, Table 1: The Table implies that PF had a temperature constant to 0.01 oC over its 4 m depth. I found this unlikely. Please justify.

Response: you are correct. We changed this value to 0.8 °C, which is certainly all that the measuring device allows.

p. 5, Lines 10-16: I don't see the point in telling us about data collected that is not analysed in the article.

We removed references to other studies and their sample material.

p. 5, Lines 15: What does "res" mean?

Typographic error: corrected.

p. 6, Sect 3.2: I think information about transport and storage of the cores is missing (e.g. temperature) and seems as important as other details that are provided.

We added "...transport in the frozen state..."

p. 6, Line 11: Define EC first time used.

Adopted.

p. 6, Line 12: "as soon as possible" is not very specific

Adopted.

p. 7, Line 2: "Slope"

Adopted.

p. 7, Line 23: "100 ppm that were"

Adopted.

p. 7, Line 27: "from the same bottle"

Adopted.

p. 8, Line 14: What is the "ice-free transect area" and why was it needed as scale in the photos?

We apologize for that error, and changed the sentence from:

"*A measurement tape at the side of the ice-free transect area served as scale in the images*"

to

"A measurement tape at the side of the cleared transect area served as scale in the images and to measure the sizes of seeps and bubble types."

which refers back to the clearing of snow from the ice transect.

p. 8, Line 22: What are "the cores of the water bodies"?

Changed to: "We compared variability of measured parameters within each ice core as a function of depth below the ice surface, and between sets of cores from each water body."

p. 8, Line 27: "identified"

Corrected.

p. 8, Line 27: How is regelation ice from snow melt identified?

Added: "...based on the appearance of the ice during sampling."

p. 8, Sect 4.1: Difficult to follow as the reader needs to keep referring to which core numbers are from which site. A sketch of the ice types would reduce the need for detailed description and give a better overall view of the structure of the ice covers.

> We have renamed the cores to be explicit about which site they come from and simplified the language in this section.

p. 9, Figure 2: Why does PF look so small on this figure? On Figure 1 no dimension of PH seems to be less than GL? In addition it would be good to label geographic location on the transects, i.e. N, S, NW etc

> We have added compass directions to the bathymetric profiles.
> PF (LG) appeared so small, because we showed the ice AND water (i.e. most of PF was occupied by ice), but since the ice was shown in white, it sort of disappeared. We have represented the ice with a light gray tone to make this clearer. We also discovered a mistake in water depth for TB (BY) and have corrected this in the figure.

p. 9, Line 2: remove "on"

> Adopted.

p. 10, section 4.2.4 & p. 15, 4.4.1: I suggest replacing "stable" with "constant". "Stable" implies "firmly fixed" and I am not sure that this is what you wish to imply. If you do mean "stable" then I think you need to justify why you expect no change under any change in circumstances.

> Adopted.

p. 10, Line 12: spelling "composition"

> Adopted.

p. 12, Fig 3: It is very interesting that PF is colder than GL. Is this because it is shallower?

> We have expanded the discussion of ice core temperature to explain the difference, adding: "In addition, temperature increased towards the bottom of the ice (Fig. 3). The bottom ice offers a protected environment with favourable conditions for microbial metabolism: relatively warm temperatures, contact with liquid water and permeable ice. The latter permits migration of gases and nutrients, similar to marine ice, where most bacteria are located in the lowest centimetres of the ice (Krembs and Engel, 2001). At LG, the bottom ice temperature decreases during the winter. This occurs because the temperature of the underlying water remains in equilibrium with a dynamic freezing point that decreases with increasing salinity when LG is cut off from Tiksi Bay. The ice surface temperature at the time of coring was primarily a function of snow cover and air temperature. The ice coring locations for LG exhibited colder ice surface temperatures and steeper gradients compared to ice coring locations at LK. Ice has a high thermal conductivity and is susceptible to quick temperature changes. Since ice temperatures were also observed for windswept areas at LK, decreasing air temperatures from 8 April 2017 (final LK coring day) to 11 April 2017 (LG coring day) could explain the generally colder ice temperature profiles at LG. "

p. 15, Fig 6 caption: "free". Has the ice free area been marked on Fig 1?

> Thank you for finding the error and for the suggestion regarding the overview figure. We also changed "ice-freed" the caption of Fig. 6 to "snow-cleared". The scale in Fig. 1 is too small to allow proper representation; the position of the bubble transect is marked in Fig. 2, to show the overlap to the ice core sites.

p. 15, Line 13-14: Is this statement tested in the present article, or is this speculation to introduce the Discussion?

> We have deleted:
> "It may act not only as a barrier, but also as a source or sink for methane or as a habitat for microbes that facilitate methane consumption."

p. 16, Line 10-11: It was not at all obvious to me how the data presented showed that the type of water body determines the circulation of methane. Please explain.

> We have re-structured and re-written parts of the abstract and conclusions, and hope that we more clearly explain how the data show differences in methane pathways from sediment to ice.

p. 16, Line 15: Suggest replacing "impact" with "setting"

> Adopted.

p. 16, Line 18: ". . .(2018), while in winter, when the connection.."

> Adopted.

p. 16, Line 26: Why would the freezing velocity be approximately constant? It is likely to decrease as 1/(ice thickness).

> The poorly worded point here was that the fractionation primarily reflects the composition of the water beneath the ice, rather than the smaller effect of shifting freezing rate. We have changed the first 2 sentences from:
>
> "*Firstly, in an open system such as TB, the water circulates freely beneath the ice cover, impeding the enrichment of lighter water isotopes in the remaining water. Therefore, the isotope composition of the initial ice should remain more or less constant, and hence also that of the ice with depth (Gibson and Prowse, 1999), assuming the freezing velocity is roughly constant.*"
>
> to:
>
> "Firstly, in an open system such as BY, the water circulates freely beneath the ice cover. The isotope composition of the ice should remain more or less constant over the winter (Gibson, 1999), and reflect the fractionation resulting from freezing of Tiksi Bay water."

p. 16, Line 24-29: The authors note that they have not taken into account the freeze fractionation influences (e.g. see Toyota et al., 2013), based on the assumption that the freezing velocity is roughly constant. If this is not necessary please provide an order of magnitude calculation to convince the reader that this is a small effect.
Toyota, T., I.J. Smith, A.J. Gough, P.J. Langhorne, G.H. Leonard, R.J. Van Hale, A.R. Mahoney, and T.G. Haskell. (2013) Oxygen-isotope fractionation during the freezing of seawater. Journal of Glaciology. Vol. 59, No. 216, 2013 doi:10.3189/2013JoG12J163

> Added text:
>
> "Oxygen isotope fractionation during the freezing of sea water has been addressed by Toyota et al., (2013), through laboratory experiments and field observations. These authors demonstrate a general dependency of increasing isotope fractionation with decreasing ice growth rate. Therefore, faster freezing induces less isotope fractionation as compared to slowly formed ice at a later stage of sea ice formation. The difference between both is within 1‰ for a large range of ice growth rates."

p. 17, Fig 7: Very nice helpful sketches. They might be useful earlier in the manuscript.

> We consider these sketches to be part of the outcome of the study, which require the explanations given in the discussion. They are based on the results, and were not a priori knowledge of the site or the processes that we would discover.

p. 18, Line 2: "alone"

> Deleted.

p. 18, Line 5: "may capture"

> Adopted.

p. 18, Line 5-7: Interesting observation that may be compared with the results of Smith et al., (2016). Smith, I.J., Eicken, H., Mahoney, A.R., Van Hale, R., Gough, A.J., Fukamachi, Y., Jones, J. (2016). Surface water mass composition changes captured by cores of Arctic land-fast sea ice. Continental Shelf Research, 118:154-164, doi:10.1016/j.csr.2016.02.008.

> Added the text:
>
> "Episodic advection of meteoric water during the winter season was also detected in land-fast sea ice cores from Barrow, Alaska (Smith et al., 2016)."

p. 18, Line 7: "indicative for the preceding freezing process (Souchez and Jouzel,1984)." I'm not sure what this means? Does this mean that freeze fractionation should be taken into account?

> Changed to: "This is reflected in the regression lines of the d18O-dD plot for BY, which differ in slope for the two sections (Fig. 8) and indicate a shift in fractionation (Souchez and Jouzel, 1984)."

p. 18, Line 9: "Tab ??". This is not at all obvious from Fig 8.

> Corrected. We have added a table of regression line statistics to show the basis for distinguishing groups of lines.

p. 18, Line 18: "but with the carbon isotope signature". I could not see much change in the carbon isotope?

Here the word "same" was missing and has been added.

p. 18, Line 29: "(Lacelle, 2011)"

Adopted.

p. 18, Line 29: Again, this is not at all obvious from Fig 8.

We have added a table of regression line statistics to show the basis for distinguishing groups of lines.

p. 20, Line 29: "was"

Adopted.

p. 21, Fig 8: The changes in slope appear to be important but cannot be seen on the plots as currently displayed. Please consider how to display this information to match the Discussion. Why is global rather than local meteoric line used?

To make the data less ambiguous, we have added a table with regression line coefficients (Table 3).

p. 22, Fig 9: Is it not possible to write down the equation of the model displayed?

The equation is in the methods section (Eq. 1). We have added a cross-reference to it in the figure caption. All parameters used for the Rayleigh model are given in the figure caption, i.e. the reader is provided with all information used to generate the plot.

p. 22, Conclusions: Please return to the aims of the study here, and show how they have been moved forward.

We have considerably changed the conclusions, please see track changes version.

p. 23, Line 8: It is not obvious to me how the data provided has shown that the ice examined "acts primarily as a barrier to methane fluxes to the atmosphere, a barrier that is effective for most of the year but also will be effected by rapid changes due to Arctic warming and associated ice thinning." Please make this clearer in the Discussion and/or Conclusion.

We have deleted this sentence.

p. 23, Line 8: What does "providing a habitat for methane oxidation" mean? Again this needs to have been explained earlier in the paper.

Changed to "...providing a habitat for methane oxidizing microorganisms." This is also described in an expanded form in the discussion on Page 19, lines 13-16:

"In addition, temperature increased towards the bottom of the ice (Fig. 4). The bottom ice offers a protected environment with favourable conditions for microbial metabolism: relatively warm temperatures, contact with liquid water and permeable ice. The latter permits migration of gases and nutrients, similar to marine ice, where most bacteria are located in the lowest 15 centimetres of the ice (Krembs and Engel, 2001). At LG, the bottom ice temperature decreases during the winter. This occurs because the temperature of the underlying water remains in equilibrium with a dynamic freezing point that decreases with increasing salinity when LG is cut off from Tiksi Bay. "

and lines 25-26:

"During freezing of the ice cover, its growth rate decreases (cf. Anderson, 1961), providing more time and space for bacterial $_{25}$ metabolism. $CH_4$ uptake from the water into the bottom of the ice and its oxidation there may have continued over the winter until the ice break-up. $CH_4$ oxidation ceases when concentrations are too low for oxidation to be efficient (Cowen et al., 2002; Valentine et al., 2001), at values ranging from 0.6 nM to 10 nM. $CH_4$ concentrations in the ice above 130 cm (Fig. 6) were less than 10 nM, suggesting that ice is an effective sink for $CH_4$ in this type of water body during winter."

---

## Author Comment (AC2) · 7 Sep 2020

Blaize Denfeld (Referee)
bdenfeld@gmail.com

This study attempts to improve our understanding of CH4 pathways in ice covered water bodies by focusing on ice and inland water continuum from lake to coast, both seldom included in the limited winter CH4 inland water studies. In doing so, the study highlights varying CH4 concentrations explained by the geomorphological differences between the aquatic systems. This is an interesting study with a unique data set and with revisions could be a nice addition to the scientific community.

**General Comments:**
1) The studies focus is on Arctic waterbodies underlain by permafrost. However, the introduction covers a broader range, i.e. when referring to Arctic and Northern lakes not all are in continuous permafrost zones. I think this broad perspective of northern lakes is appropriate but think the introduction could be better structured to go from a broad approach (Northern ice-covered lakes) to lakes in a continuous permafrost landscape. In particulate Paragraph 2 of the introduction would benefit from this restructuring.

> We have substantially re-written the introduction, slimming it down, and removing the tacit differentiation between ebullition and diffusive fluxes of methane.

2) Additional details needed to be clarified in the methods, particularly on CH4 lab sampling. See specific comments below.

> Answered below.

3) The discussion is sorted into the three different water bodies, which works but it would be helpful to also have an overview of how the values calculated for these water bodies compare with values in the same system (lakes/coastal permafrost areas). Perhaps reporting a range for all 11 ice cores and discussing how it compares to other ice-covered inland water values. This could be done in the initial discussion paragraph before diving into the specifics of the three water body types.

> We compare ranges in the results table but have not added comparisons to other values. These are simply not available for all three water body types covered in this paper, nor are their seasonal variabilities characterized. This is part of what makes our paper unique.

4) In the conclusion, in addition to returning to the aims of the study it would be nice to know how these findings fit into understanding the lake-lagoon-coast transition in the arctic region. Although I appreciate that caution should be taken in making large upscaling statements given the limited sample size and snapshot in time.

> We are trying to strike a balance between restricting ourselves to the conclusions permitted by our results, and explicit statements of what we feel are the implications of these findings.

5) Much of cited literature focuses on findings from lakes. Are there any studies that have looked at CH4 concentrations below ice in coastal areas? If so, it would strength the paper to include them in the introduction and discussion (see general point 3).

> We agree, but are not aware of any additional studies from coastal ice.

**Specific Comments:**
Title: Since the study only investigates one lake, lagoon and bay perhaps the title would better represent the actually study as, "Methane Pathways in Winter Ice of Thermokarst Lake-Lagoon-Coastal Water Transect in North Siberia"

> Yes, corrected.

P1, L 7-9: Could use a tie in sentence and possible move this information after L 5-6 as it is continuing to point out differences between the system, e.g "In addition the three water bodies had different freezing systems. In TB. . ."

> We have moved the sentence.

P1, L 12: is "above the ice-water interface" referring to in the ice? If so please clarify.

> Clarified by including "in the ice".

P 2, L 7-9: This sentence should be rewritten. Is the idea that $CH_4$ can continue to accumulate in lakes over the ice-covered period whereas in soils the active layer freezes during winter and $CH_4$ is not produced? A reference showing that $CH_4$ production in the active layer of permafrost is mainly during summer would help support this statement. $CH_4$ has been found to accumulate in shallow lakes over winter, so the authors may need to think about the definition of "certain circumstances".

> As part of revision of the introduction, these sentences have been removed. We do not seek to compare aquatic and soil environments.

P2, L 23-25: I assume these two sentences are referring to lake sediments? Please clarify.

> This sentence has also been deleted.

P2, L 25: a third pathway, plant mediation, should be included.

> This sentence has also been deleted.

P3, L 1-4: Are these the only two studies looking at $CO_2$ and $CH_4$ in ice? Perhaps these sentences could be simplified as one, "Of the limited studies, accumulation of $CH_4$ in and under the ice during winter were realized for shallow ice-covered lakes in Alaska (Phelps et al. 1998) and four lakes in discontinuous permafrost area (Boereboom et al. 2012)." Or something like that.

> Adopted.

Pg 3, L6, "However, methane oxidation. . ."

> Adopted.

P3, L10: change to, "methane has been found to oxidize at temperatures as low as "C."

> Adopted.

**Material and Methods:**

P5, L 15: typo "res"

> Corrected.

P6, L 1-9: How were the ice samples stored, in -15C? Could there have been $CH_4$ oxidation? How effective was the vacuum pump at removing $O_2$?

> We describe that the ice cores were stored in sealed plastic tubes and in thermally insulated boxes for transport. We have added: "..., which effectively sealed the inner bag surface to the ice." We did look at our data critically and consider potential or plausible roles played by oxidation after sampling. The observed methane concentrations and isotope concentrations are not consistent with oxidation during transport.

Pg 6, L 22-25: How many samples were considered high salinity and low salinity?

> Now specified in text (7 were "high").

Pg 7, L 14: Remove Global Meteoric Water Line, already abbreviated above.

> Adopted.

Pg 7, L 19-20: More details needed here. When the $N_2$ was added did it create an overpressure in the vial or was 5 mL of water removed? How was the water sample equilibrated with the $N_2$, shaken? Was the equilibrated air then removed from the vial and injected into the GC?

> Added: "For methane concentration, 5 ml $N_2$ was added and 5 ml of water simultaneously removed from the vials. Afterwards, the water sample was equilibrated with the $N_2$ by being shaken for 1 hour at room temperature. Then, 1.5 ml of the equilibrated air was removed from the vial and injected into a gas chromatograph (GC; Agilent 8900) with a flame ionization detector (FID)."

Pg 8, 10-19: Was the bubble transect done before or after the ice coring, i.e. were the ice core samples taken from the targeted plot area or were they randomly selected?

> The bubble transect and coring happened during the same time period but were independent of each other, i.e. cores were not taken from the bubble transect.

P 18, L 9: typo, "(Tab. ??)"

> Corrected.

P 18, L 10: typo, "admixture"

> Corrected.

P 19, L 12-16: Could it also be because less is being produced?

> This would not explain the shift in isotope composition.

P 20, L 29: typo, "lwas"

> Corrected.

Table 1: In the legend add "Water" at the start of the second sentence. For the electrical conductivity column could you report Salinity [PSU] /EC [mS/cm] since salinity is known for all three water bodies. Or have two sperate columns for salinity and EC.

> We have created two columns with a conversion to salinity.

Table 2: For PF replace Lake with Lagoon in the Sampling Site header.

> Corrected.

Figure 4: GL has different scales for the first two rows

> This is necessary, since the values for one core (LK-5) are so extreme and using the same scale would mask any variation at the other 2 sites. We have added to caption: "Note that the isotope concentration scales for Goltsovoye Lake (LK, at right) differ from the other 2 sites to accommodate the values observed in core LK-5."

Technical Comments:

1) There are many places where Methane is written out, since it is abbreviated to CH4 on P1 L19 it should be changed to CH4 thereafter.

> Corrected, also in figure captions.

2) Keywords: remove . . . at end of list

> Corrected.

---

## Author Comment (AC3) · 7 Sep 2020

**Compare Results**

| Old File: | | New File: |
|---|---|---|
| **tc-2019-304-manuscript.pdf** | versus | **tc-2019-304-manuscript_revised_20200907.pdf** |
| **28 pages (16.65 MB)** | | **30 pages (24.14 MB)** |
| 2020-02-19 3:03:33 PM | | 2020-09-07 8:39:30 PM |

**Total Changes**

**525**

Text only comparison

**Content**

307 Replacements

112 Insertions

106 Deletions

**Styling and Annotations**

0 Styling

0 Annotations

Go to First Change (page 1)

[revised manuscript text omitted]

---

## Author Response (AR2)

**Editor Decision: Publish subject to minor revisions (review by editor)** (13 Dec 2020) by Petra Heil
Comments to the Author:
tc-2019-304 Submitted on 12 Dec 2019
Methane Pathways in Winter Ice of Thermokarst Lake-Lagoon-Coastal Water Transect in North Siberia

Ines Spangenberg, Pier Paul Overduin, Ellen Damm, Ingeborg Bussmann, Hanno Meyer, Susanne Liebner, Michael Angelopoulos, Boris K. Biskaborn, Mikhail N. Grigoriev, and Guido Grosse

Editor's comments:
==================
Dear Ines, Paul and co-authors.

Thank you for providing a much improved ms to review. This revised version is much easier to follow.

General comments:
* Pls use SI units. For example: 5-14: Change "10 cm" to 0.1 m".
**In most cases we have adopted this requirement, including in all tables and figures. For electrical conductivity, "µS/cm" are almost universally accepted, especially for snow and ice. To avoid rendering our data obscure, we have left these as is.**
* All measurements, incl for example in 3.3 Hydrochemistry in ice:
Include full specifications or error characteristics for each sensor. **Done**
* Pls include some information and discussion on the statistical methods used in the manuscript.
**See specific comments below.**
* The prevalence and impacts of snowloading on the ice should be discussed in some more detail (p20). Can you speculate on this process in a changing climate?
**Recent changes to solid precipitation in the Lena Delta suggest that it has been increasing over the past 20 years, and it is projected to do so; since we did not track the distribution of snow over the winter, only observed its thickness at the time and location of coring, and since we do not have a baseline of observations at this location, it is somewhat difficult to speculate using the results we show here on the effects of increased snow load. All coring locations are subject to snow redistribution by wind over the winter. In our study snow loading that resulted in snow integrated in the floating ice was only observed at the eroding bluff of the thermokarst lake, where the bluff acted as a wind-leeward trap for snow.**
* Rewrite the Conclusions section to make more impactful.
**We have changed the conclusions from:**
**"CH4 concentrations in the seasonal ice cover of three types of Arctic water bodies, representing three different stages of permafrost degradation, revealed differences related to the process of ice formation and its importance as mitigator of CH4 fluxes to the atmosphere. In the ice of Tiksi Bay, open to the central Laptev Sea throughout the winter and underlain by permafrost, the signatures of the stable isotopes of water and electrical conductivity reflected the composition of the upper layer of brackish water throughout the winter, with an increasing proportion riverine waters during winter. In this setting, CH4 concentrations were low but, as in all three water bodies, supersaturated with respect to atmospheric concentration.In the coastal Polar Fox Lagoon, a breached thermokarst lake, ice formation sealed the channel between the lagoon to the sea.This isolated and concentrated the remaining brackish water beneath the thickening ice during the winter. CH4 was present at variable concentrations, but the concentration profile over depth and the stable isotope signatures strongly suggest that bacterial oxidation takes place at the interface**

[revised manuscript text omitted]

Minor comments:

Throughout manuscript:

Pls change "e.g." to "e.g.," and "i.e." to "i.e.,". **Changed**

2-10: Remove "Generally, ". **Done**

2-17: "Escape" from where? -- I would also prefer to be specific and refer to "ice- and snow-free environments" rather than "in summer".

**Changed to: " While gas may easily escape from thermokarst lakes to the atmosphere in ice- and snow-free periods, an ice cover forms a barrier for 9 to 10 months in winter. During the wintertime, gas bubbles are trapped under and eventually within the ice."**

3-16: Instead of "71° 40' - 71° 80' N and 129° 00' - 129° 30'E" decimal lat and lon would be more contemporary. **Corrected**

3-27: Change "less than 11 m in general" to "largely less than 11 m deep". **Changed**

3-28: Change "is located southeast" to "is located to the southeast". **Changed**

3-28: Remove "the" from "the Bykovskaya Channel". **Removed**

3-33: Correct "can be disturbed by storm events" to "may be disturbed by storm events". **Changed**

4-Cap1: What does "(c)" in "((c) DigitalGlobe)." mean? **Corrected**

4-1: Correct "Tidally-based sea-level oscillations" to "Sea-level oscillations driven by tides" **Changed**

4-7: Would you turn "Tab. 1 lists characteristics of the studied water bodies." into an active statement about the water bodies or their characteristics please? **Changed**

5-3: Correct "cores were drilled" to "cores were recovered". **Changed**

5-5: Pls make the statement "Tab. 1 lists the mean ice thicknesses of the sampled ice core for the locations." an active one about the ice cores and their thickness. **Changed**

5-8: Correct "temperature was measured" to "vertical temperature profiles were obtained". **Corrected**

5-8: Correct "every 10 cm," to "every 0.1 m,". **Corrected**

5-12: Can you pls provide the specifications of the sensors used in this study? E.g., resolution and accuracy of the CTD sensors? **Added "The accuracy and resolution of the devices were ±0.05 and 0.01 ∘C, respectively, for temperature and ±5 and 1 µS/cm−1 for electrical conductivity. "**

5-15: Change "(over 1-2 days)." to "for 1 -- 2 days." **Changed**

6-23: Correct "An δ 18 O-δD plot gives" to "The comparison of δ 18 O to -δD provides". **Corrected**

6-26: Clarify "equilibrium conditions": Equilibrium of what? **changed to "equilibrium freezing conditions."**

6-29: Change "for the first ice" to "for new ice". **As it stands, "first ice" refers to the first ice formed and not to new ice or ice that formed at a later date - this is an important distinction.**

7-7: Correct "kept cold" to "kept cool". **Corrected**

8-7: Correct "The photos were rectified" to "The images were orthorectified". **Corrected**

8-11: Provide info on "R environment". --> I.e., that is is a software based on xxx to do yyy or similar. See your info on AGISoft. --

**Added "(a free software environment with interpreted computer language for statistical computing and graphics, www.r-project.org)"**

Provide info on the 14 distance classes and how they affect the kernel calculations. (So there are no surprises to the reader when "class A seeps" etc. are noted further down in the ms.)

**We agree and added information on how we calculated 14 distance classes. The classes do not affect the density function that is based on the original data distribution. To explain better we added the following text: "We chose 14 distance classes in a Kernel density estimation guided by two criteria: 1) only allow bin size in which each bin is represented by seep data, and 2) maximize visualization of the density trend over the profile."**

8-15: Remove "densely". **Removed.**

8-19: Correct "was identified" to "were identified". **Corrected**

8-25: Correct "until the depth of about 112 to 114 cm." to "down to a depth of ca 1.12 to 1.14 m." **Corrected**

8-25: What is a "solid" snow cover. This is not a technical term. Change to something like "The snow cover generally had a hard surface and was characterized...".

**Changed to: "The snow cover was hard-packed and characterized by different melt forms."**

8-26: Rewrite "thickness of the snow-layer ranged from 0 cm": The thickness cannot be Zero. Instead there is an absence of snow.

**Changed to: "Snow was either absent (LK-2 and LK-3) or 0.15 m (LK-4), 0.23 m (LK-1) and 0.92 m (LK-5) thick."**

8-26: Change "For all cores, no algae inclusions were visible (Strauss et al., 2018)." to "In none of the cores algae inclusions were detected (Strauss et al., 2018)." **Changed**

9-Fig2: Explain the blue and grey shading. I assume they are for the water and the ice. - It appears that the ice thickness is constant for each profile. Pls correct.

**That is correct. The grey colour shows the mean ice thickness for each water body, since we did not measure ice thickness between boreholes. Changed caption to: "Figure 2. Cross-sections of the bathymetry of the Tiksi Bay (BY) profile (N to S), Polar Fox Lagoon (LG) from southwest to northeast, and Goltsovoye Lake (LK) along the coring transect (W to E). Positions of the ice cores are indicated as numbered vertical lines in the ice layer (grey shading shows mean water body ice thickness) and the water column below the ice is indicated in blue. The position of the bubble transect at Goltsovoye Lake is represented with a dashed red line."**

12-Tab2: Rename "site" or "sampling site" to "transect". **Changed**

12-7: Add close bracket to read "(Fig. 5)." **Changed**

14-14: Remove "values" from "values ranged". **Changed**

14-14: Change "The values" to "These values". **Changed**

14-14: Correct "smaller" to "lower". **Corrected**

14-15: Correct "for the other" to "of the other". **Corrected to "of the cores of the other locations"**

14-15: Remove "values" twice and "the" from "In LG, the". **Removed**

14-17: Remove "The" and "values". **Removed**

14-17: Change "The pattern is similar but inverse to the CH 4 concentrations." to "The pattern is inverse to that of the CH 4 concentrations." **Changed**

14-20: Correct "is quite constant." to "is relative constant." **Corrected**

14-25: Change "A seasonal ice cover is" to "The seasonal ice cover forms". **Changed**

14-26: Correct "covered by ice for 9 months of the year" to "typically ice covered for about 9 months every year". **Corrected**

14-28: Correct "all increase," to "all increased,". **Corrected**

14-30: Correct "shortens" to "will shorten". **Corrected**

16-21: Correct "whens" to "when". **Corrected**

17-Fig7: Change "studied in this paper" to "under investigation". **Changed**

18-26: Correct brackets: One more closing than opening brackets. **Changed to "(4.8 m to 8.3 m below the sediment surface, Angelopoulos et al., 2020)".**

18-27/28: Not a sentence. Pls rewrite. **Changed to "For ice that formed before the lagoon was separated from the sea (above 0.6 m), the isotopic signature indicates freezing under equilibrium conditions (Lacelle, 2011), with a slope of 8.2 between δ18O and δD (Fig. 8, Tab. 3)."**

18-last para: The use of "later (deeper)" and "earlier (upper)" and similar is confusing. Pls rewrite.

**We have simplified by removing references to "earlier/later" ice and by referencing to ice core position (upper vs lower, relative to specific depth).**

19-6: Use SI units: "60 cm". -- Throughout manuscript. **Changed**

19-31: Correct "euqilibrium" to "equilibrium". **Changed**

20-12: Replace "This circumstance clearly shows" with "Our data demonstrate". **Changed**

20-16: Rewrite "involvement of snow". **Changed to "indicate snow as a source".**

20-32: Correct "Fi. 6" to "Fig. 6". **Corrected**

29-22: Should "Walter Anthony, K.," read "Walter Anthony, K.M.,"? - not least for consistency?
**We agree, but this is not how the different journals have cited the same author. Katey's surname has changed with marriage and both are used in citations, and some journals neglect to use her middle initial. To make the papers findable, we follow the citation provided by the journals.**

References: Not checked, nor cross searched.

For final publication, the manuscript should be
**accepted as is**
accepted subject to **technical corrections**
accepted subject to **minor revisions**
reconsidered after **major revisions**
 I am willing to review the revised paper.
 I am **not** willing to review the revised paper.
**rejected**

**Suggestions for revision or reasons for rejection (will be published if the paper is accepted for final publication)**
This paper describes ice on three distinct water bodies, in particular examining the methane within the ice and the physical properties associated with understanding the observed methane concentrations. The authors have taken considerable care to respond to the comments of the reviewers and (in my opinion) the revised version is much improved. The authors are much clearer about the aims of their study, and the description of the science and conclusions is clarified. The paper is now easy to read and almost without typographic error.

As before I alert the editor to the fact that I am not expert on the suite of chemical techniques involved in the study.

I have a few remaining minor comments.

Comment 1: The title of the study (and I do prefer the new title) and the focus of the Introduction and Discussion is on winter ice. However the ice is sampled in spring. I would like some discussion regarding why the authors believe that the measurements are representative of winter. For example is the sampled ice thickness close to the maximum seasonal ice thickness? Is the ice still growing at the time of sampling? For those readers not familiar with the progress of the seasons in this geographic area, it would be useful to relate the seasons to the dates of the year. It would be very useful if the dates of sampling were given in Table 1.
**Added:**
**"Ice growth ceases when heat flux to the atmosphere slows, and is negligible by the end of April at our study site. Ice was cored at close to its maximum thickness to include almost the entire winter record of freezing. Ice cores were drilled on April 8, 2017 (LK), April 10, 2017 (LG) and April 11, 2017 (BY)."**

Comment 2: I would still like more information about transport and storage of the cores (for example on p. 5, Sect 3.2). What was the approximate temperature during transport; how long between taking cores and performing the analyses? It is important for the reader to be convinced that there have not been irreversible changes during transport.
**Added:**
**"Cores were stored in freezers after drilling, in a permafrost tunnel (*lednik*) while waiting for transport and then transported by refrigerated truck at freezer temperatures (-18°C). We did not record temperature during transport, but based on a comparison of photos of the cores after drilling and in the laboratory, the structure and ice morphology of the cores were preserved during transport. We are therefore confident that not even surficial melting took place. "**
**We had included the timing of sampling and analyses, but the other reviewer asked that it be removed.**

Comment 3: Could snow cover be added to Fig. 2 to help the reader understand the temperature measurements and the snow loading described in section 5.3?

**Snow cover is in general not thick enough to be distinguishable in Figure 2, we do not have data beyond depths at drilling locations and snow is redistributed throughout the winter. We fell that portraying the thickness at the time of sampling would be misleading. The importance of snow at our sites is captured in the ice record as described in the discussion. We therefore prefer not to include it in Fig. 2.**

Technical Corrections
p. 1: Abstract: Abstract is now much clearer.
**Thank you.**

p. 5, Table 1: In relation to statement that winter ice, please could you put dates of sampling in the Table.
**Dates have been added to Section 3.2**

p. 5, Sect 3.1: We are told the ice was sampled between Apr 5 and 12. But how is April related to winter ice?
**See above added text.**

p. 6, Line 32: "prior to freezing" or "during freezing"?
**Changed from "changes prior to freezing" to "changes as freezing progresses."**

p. 7, Line 7: "2 months between sampling and measurement" Thank you – this is the sort of information that I consider important. But I was not sure whether "sampling" meant taking the core or filling the glass bottles.
**Replaced "sampling" with "filling the sample bottles".**

p. 7, Line 9: replace "shaken" with "shaking"
**Done**

p. 8, Line 4: Fig. 2 (rather than Fig. 3). Fig. 3 is cited before Fig. 2.
**Changed to "(Fig. 2 & 3)".**

p. 8, Line 19-21: Paragraph break seems to have been placed part way through the description of BY ice. Please check.
**Checked and changed**

p. 8, Line 20: Begin sentence "On the ice of Tiksi Bay, the snow thickness…"
**Done**

p. 12, Lines 1-2: A rather strange sentence "While the …. (below 80-90 cm)" I think this sentence is unnecessary but I leave this as a decision of the authors.
**We have deleted the sentence.**

p. 12, Line 7: "(Fig. 5)"
**Changed**

p. 14, Lines 7-9: This is a repetition of Lines 5-7.
**Deleted**

p. 14, Lines 19-21: This could be written more concisely and without repetition.
**Changed from:**
**"In LK, the δ13C-CH4 values range from −91.6 to −12.3 ‰ (Tab. 2). The highest and lowest values occurred in LK-3 and LK-5. These two cores show changes in the δ13C-CH4 values with depths, whereas the stable carbon isotopic signal of the other cores (LK-1, LK-2 and LK-4) varies between −46.8 to −43.3 ‰ and is quite 20 constant. In LK-1, LK-2 and LK-4, the δ13C-CH4 values had a mean of −43.3 ‰ and were uniform (±2.2 ‰) with depth, in**
**contrast to LK-3 and LK-5, where values ranged from −91.6 to −12.3 ‰, with a strong variability within and between the two cores. Greater variability was observed for CH 4 concentrations."**
**to:**
**"In LK-1, LK-2 and LK-4, the δ13C-CH4 values had a mean of −43.3 ‰ and were uniform (±2.2 ‰) with depth, whereas values in LK-3 and LK-5 ranged from −91.6 to −12.3 ‰, with a strong variability within and between the two cores (Tab. 2). A greater variability was observed for CH 4 concentrations."**

p. 14, Lines 25-26: Please tell us which months and how these months relate to winter and spring (see Comment 1).
**See above text.**

p. 16, Line 6: Please tell us when the onset of ice formation took place on BY.
**Fall cloud cover and storm events make it difficult to say when the lasting ice cover initiated at any location.**

p. 16, Line 7: Please mark Muostakh Island on Fig. 1.
**Muostakh Island is not shown on Fig. 1. Changed to "...south of Cape Muostakh.", which is inclusive.**

p. 16, Line 21: delete "s" from "whens"
**Done.**

p. 16, Line 21-22: "The difference between both is within 1‰ for a large range of ice growth rates." I don't know what is meant by "both". Please clarify.
**This sentence has been deleted. The previous sentence makes the actual point.**

p. 16, Line 35: I could not find a reference to Fig. 7 in the text. I think Fig. 8 is referred to before Fig. 7.
**Corrected.**

p. 18, Line 1-4: Why should there be mixing of saline water with additional water of meteoric origin when the ice was 90 cm thick? At what approximate time of year did this occur? Why would it take place if additional river outflow was unlikely at that time? Please speculate on a physical reason for your observations.
**Added: "Lena River water from much further south is carried into Tiksi Bay throughout the year and plots close to the Local Meteoric Water Line (slope 7.3, Juhls et al., 2020). In April this flux is still at base flow levels but contributes to the least dense surface water layer beneath the ice. "**

p. 18, Line 26: Formatting of Angelopoulos et al

**Done**.

p. 19, Line 20-23: "Ice has a high thermal conductivity and is susceptible to quick temperature changes. Since ice temperatures were also observed for windswept areas at LK, decreasing air temperatures from 8 April 2017 (final LK coring day) to 11 April 2017 (LG coring day) explain the generally colder ice temperature profiles at LG."
I am not convinced by this explanation because the temperature of the ice at depths greater than 100 cm are higher for LK than LG and the air temperature takes some time to propagate to this depth. I suspect that the thickness of the snow cover is much more important since ice has a higher thermal conductivity than snow. More detail on dates of sampling (Comment 1) and a description of the snow cover (Comment 3) may help explain these observations.
**Please consider the primary argument that we make, which is that the water in LG is colder than in LK, due to cooling below 0°C as freezing progresses. This most likely drives temperatures deeper in the ice, for example at 100 cm as you point out, which reflect conditions prior to drilling, not air temperature changes between drilling dates. The snow thickness on LK ranged from 0 to 0.92 m, but all of the ice temperature profiles are grouped closely together when compared to LG (0.08 to 0.23 m), so that the water body, rather than the local snow thickness, is dominant in explaining the observed temperature profiles. We have added all of the snow thicknesses to the ice morphology description.**

p. 19, Line 31: "equilibrium"
**Changed.**

p. 19, Line 31 & p. 20, Line 14-30: Snow loading discussion is very interesting. Please see Comment 3.
**Please see above answer. We must restrict ourselves to the evidence of snow loading and inclusion in the ice record.**

p. 23, Fig 9: I am surprised that the dashed line is the fit to the plus markers. This figure might be clearer and the fit more obvious if you plotted a linear fit of (™13CCH4-(™13CCH4)0) versus lnf
**To be clear, the line is not a fit to the symbols nor an indication of correlation. The lines show the modelled Rayleigh fractionation using only the values for initial d13C-CH4 and the alpha value described in the text. To make this clearer, we have increased the length of the modelled values to lower fractions and changed the caption to: "Figure 9. Observed δ13C-CH4 and CH4 concentrations in the ice of Polar Fox Lagoon (LG) for shallow and deep ice (above and below 0.6 m, respectively, shown as symbols). The lines show δ13C-CH4 calculated based on Rayleigh fractionation during oxidation (Eq. 1). For these modelled values, α was set to 1.004 and the initial δ13C-CH4 values were 195 nM (0 to 0.6 m, dotted line) and 450 nM (>0.6 m, solid line) with initial isotopic signatures of −80‰ and−70‰, respectively."**

---

## Author Response (AR3)

**Editor Decision: Publish subject to technical corrections** (15 Feb 2021) by Petra Heil
Comments to the Author:
Editor's comments:
==================
Dear Paul and co-authors.

Thank you for working through the comments and providing a much improved ms.
I congratulate you to proceed this manuscript through the TC peer-review process.

Some MINOR Comments/Suggestions:

5-14: Replace "to Germany" with "to the analytical/mechanical/.. refridgerated laboratory" or similar.
6-1: Could remove "in Potsdam".
9-5: Rewrite "In none of the cores algae inclusions were detected [REF]." to "None of the cores contained algae inclusions [REF]." (or similar)
12-Fig4: Y-axis labels: Capitalize "depth" to read "Depth". 3 times
14-Fig5: Y-axis labels: Capitalize "depth" to read "Depth". 3 times
15-Fig6: Y-axis labels: Capitalize "depth" to read "Depth". Twice.
24-15: Suggest to acknowledge the reviewers of this review processes for their input.

**We have adopted all suggested changes.**